# MEMORY-ENHANCED NEURAL SOLVERS FOR EFFICIENT ADAPTATION IN COMBINATORIAL OPTIMIZATION

## ABSTRACT

Combinatorial Optimization is crucial to numerous real-world applications, yet still presents challenges due to its (NP-)hard nature. Amongst existing approaches, heuristics often offer the best trade-off between quality and scalability, making them suitable for industrial use. While Reinforcement Learning (RL) offers a flexible framework for designing heuristics, its adoption over handcrafted heuristics remains incomplete within industrial solvers. Existing learned methods still lack the ability to adapt to specific instances and fully leverage the available computational budget. The current best methods either rely on a collection of pre-trained policies, or on data-inefficient fine-tuning; hence failing to fully utilize newly available information within the constraints of the budget. In response, we present MEMENTO, an approach that leverages memory to improve the adaptation of neural solvers at inference time. MEMENTO enables updating the action distribution dynamically based on the outcome of previous decisions. We validate its effectiveness on benchmark problems, in particular Traveling Salesman and Capacitated Vehicle Routing, demonstrating its superiority over tree-search and policy-gradient fine-tuning; and showing it can be zero-shot combined with diversity-based solvers. We successfully train all RL auto-regressive solvers on large instances, and show that MEMENTO can scale and is data-efficient. Overall, MEMENTO enables to push the state-of-the-art on 11 out of 12 evaluated tasks.

## 1 INTRODUCTION

Combinatorial Optimization (CO) encompasses a vast range of real-world applications, ranging from transportation (J-F Audy & Rousseau, 2011) and logistics (Dincbas et al., 1992) to energy management (Froger et al., 2016). These problems involve finding optimal orderings, labels, or subsets of discrete sets to optimize given objective functions. Real-world CO problems are typically NP-hard with a solution space growing exponentially with the problem size, making it intractable to find the optimal solution. Hence, industrial solvers rely on sophisticated heuristic approaches to solve them in practice. Reinforcement Learning (RL) provides a versatile framework for learning such heuristics and has demonstrated remarkable success in tackling CO tasks (Mazyavkina et al., 2021).

Traditionally, RL methods train policies to incrementally construct solutions. However, achieving optimality in a single construction attempt for NP-hard problems is impractical. Therefore, pre-trained policies are often combined with search procedures. The literature introduces a range of these procedures from stochastic sampling (Kool et al., 2019; Kwon et al., 2020; Grinsztajn et al., 2023), beam search (Steinbiss et al., 1994; Choo et al., 2022), Monte Carlo Tree Search (MCTS) (Browne et al., 2012) to online fine-tuning (Bello et al., 2016; Hottung et al., 2022) and searching a space of diverse pre-trained policies (Chalumeau et al., 2023b). One popular online fine-tuning strategy, Efficient Active Search (EAS) (Hottung et al., 2022), uses transitions from generated solutions to derive policy gradient updates. However, it suffers from the inherent drawbacks of back-propagation, in particular having each data point only impacting the update as much as what is enabled by the learning rate. The leading strategy, COMPASS (Chalumeau et al., 2023b), relies on a space of diverse pre-trained policies for fast adaptation, but is limited to selecting the most appropriate one, and lacks an update mechanism using the collected data. Meanwhile, the use of memory has grown in modern deep learning, demonstrated by the success of retrieval augmented approaches in Natural Language Processing (Lewis et al., 2020), and to a smaller extent, its use in RL (Humphreys et al., 2022). These

mechanisms create a closer link between collected experience and policy update, making them a promising candidate to improve adaptation.

In this vein, we introduce MEMENTO, a method to efficiently update the action distribution of neural solvers online, leveraging a memory of recently collected data. Importantly, MEMENTO is model agnostic and can be applied to many existing neural solvers. MEMENTO is able to learn expressive update rules, which prove experimentally to outperform classic policy gradient updates. We evaluate our method on two popular routing problems - Traveling Salesman (TSP) and Capacitated Vehicle Routing (CVRP). We evaluate all methods both in and out of distribution, and tackle instances up to size 500 to better understand their scaling properties. We experimentally show that MEMENTO can be combined with existing methods and provide systematic improvements, reaching state-of-the-art on 11 out of 12 tasks. We provide an analysis of the update rules learned by MEMENTO, examining how it enables to adapt faster than policy gradient methods like EAS.

Our contributions come as follows: **(i)** We introduce MEMENTO, a flexible and scalable method – composed of a memory and a processing module – enabling efficient adaptation of policies at inference time. **(ii)** We provide experimental evidence that MEMENTO can be combined with existing approaches to boost their performance in and out of distribution, even for large instances and unseen solvers. **(iii)** Whilst doing so, we train and evaluate leading construction methods on TSP and CVRP instances of size 500 solely with RL, proving to outperform all existing RL methods, and release the checkpoints. **(iv)** We open-source the implementation of MEMENTO in JAX (Bradbury et al., 2018), along with test sets and checkpoints to facilitate future research advancements.

## 2 RELATED WORK

**Construction methods for CO** Construction approaches in RL for CO refer to methods that incrementally build a solution one action after the other. Hopfield & Tank (1985) was the first to use a neural network to solve the TSP, followed by Bello et al. (2016) and Deudon et al. (2018) who respectively added a reinforcement learning loss and an attention-based encoder. These works were further extended by Kool et al. (2019) and Kwon et al. (2020) to use a general transformer architecture (Vaswani et al., 2017), which has become the standard model choice that we also leverage in this paper. These works have given rise to several variants, improving the architecture (Xin et al., 2021; Luo et al., 2023) or the loss (Kim et al., 2021; 2022; Drakulic et al., 2023; Sun et al., 2024). These improvements are orthogonal to our work and could a priori be combined with our method. It is to be noted that construction approaches are not restricted to routing problems: numerous works have tackled various CO problems, especially on graphs, like Maximum Cut (Dai et al., 2017; Barrett et al., 2020), or Job Shop Scheduling Problem (JSSP) (Zhang et al., 2020; Park et al., 2021).

Construction methods make use of a predetermined compute budget to generate one valid solution for a problem instance, depending on the number of necessary action steps. In practice, it is common to have a greater, fixed compute budget to solve the problem at hand such that several trials can be attempted on the same instance. However, continuously rolling out the same learned policy is inefficient as (i) the generated solutions lack diversity (Grinsztajn et al., 2023) and (ii) the information gathered from previous rollouts is not utilized. How to efficiently leverage this extra budget has drawn some attention recently. We present the two main types of approaches that augment RL construction methods with no problem-specific knowledge (hence excluding *solution improvement methods*).

**Diversity-based methods** The first type of approach focuses on improving the diversity of the generated solutions (Kwon et al., 2020; Grinsztajn et al., 2023; Chalumeau et al., 2023b; Hottung et al., 2024). POMO (Kwon et al., 2020) makes use of different starting points to enable the same policy to generate diverse candidates. Poppy (Grinsztajn et al., 2023) leverages a population of agents with a loss targeted at specialization on sub-distribution of instances. It was extended in Chalumeau et al. (2023b) and Hottung et al. (2024), respectively replacing the population with a continuous latent space (COMPASS) or a discrete context vector (PolyNet).

**Policy improvement at inference time** The second category of methods, which includes ME-MENTO, addresses the improvement aspect. These methods are theoretically orthogonal and can be combined with those mentioned earlier. EAS (Hottung et al., 2022) employs a parameter-efficient fine-tuning approach on the test instances, whereas SGBS (Choo et al., 2022) enhances this strategy

by incorporating tree search. While these two methods demonstrate impressive performance, they rely on rigid, handcrafted improvement and search procedures, which may not be optimal for diverse problem domains or computational budgets.

In contrast, MOCO, FER and MOCO (Dernedde et al., 2024; Jingwen et al., 2023; I. Garmendia et al., 2024) learn the policy improvement update. MOCO (Dernedde et al., 2024) introduces a meta-optimizer that learns to calculate flexible parameter updates based on the reinforce gradient, remaining budget, and best solutions discovered so far. MARCO (I. Garmendia et al., 2024) aggregates graph edges' features and introduce problem-dependent similarity metrics to improve exploration of the solution space. Both approaches enable to learn adaptation strategies but are tied to heatmap-based policies, preventing their applicability to certain CO problems like CVRP. FER (Jingwen et al., 2023) learns a rule to update the instance nodes' embedding, whereas MEMENTO directly updates the action logits, hence being architecture-agnostic, scaling better and achieving higher performance.

## 3 METHODS

### 3.1 PRELIMINARIES

**Formulation** A CO problem can be represented as a Markov Decision Process (MDP) denoted by $M = (S, A, R, T, H)$. This formulation encompasses the state space $S$ with states $s_i \in S$, the action space $A$ with actions $a_i \in A$, the reward function $R(r|s, a)$, the transition function $T(s_{i+1}|s_i, a_i)$, and the horizon $H$ indicating the episode duration. Here, the state of a problem instance is portrayed as the sequence of actions taken within that instance, where the subsequent state $s_{t+1}$ is determined by applying the chosen action $a_t$ to the current state $s_t$. An agent is introduced in the MDP to engage with the problem and seek solutions by learning a policy $\pi(a|s)$. The standard objective considered in RL works is the single shot optimisation, i.e. finding a policy that generates a trajectory $\tau$ which maximises the collected reward: $\pi^* = \arg\max_\pi \mathbb{E}_{\tau \sim \pi}[R(\tau)]$, where $R(\tau) = \sum_{t=0}^{H} R(s_t, a_t)$.

The specificity of most practical cases in RL for CO is that the policy is given a budget of shots $B$ to find the best possible solution, rather than a single shot. Consequently, the learning objective should rather be : $\pi^* = \arg\max_\pi \mathbb{E}_{\tau_i \sim \pi}[\max_{i=1,...,B} R(\tau_i)]$. It is hence beneficial to learn policies with higher entropy to provide better exploration, and ideally, policies that can use their previous attempts to (i) explore decisions they have not tried yet (ii) put more focus on promising regions of the space.

### 3.2 MEMENTO

Adaptation to unseen instances is crucial for neural solvers. Even when evaluated in the distribution they were trained on, neural solvers are not expected to provide the optimal solution on the first shot, due to the NP-hardness of the problems tackled. Making clever use of the available compute budget for efficient online adaptation is key to performance.

A compelling approach is to store all past attempts in a memory which can be leveraged for subsequent trajectories. This ensures that no information is lost and that promising trajectories can be used more than once. There are many ways of implementing such framework depending on how the information is stored, retrieved, and used in the policy. We would like the memory-based update mechanism to be (i) learnable (how to use past trajectories to craft better trajectories should be learnt instead of harcoded) (ii) light-weight to not compromise inference time unduly (iii) agnostic to the underlying model architecture (iv) able to leverage pre-trained memory-less policies.

**Overview** To this end, we introduce MEMENTO, a method that dynamically adapts the action distribution of the neural solver using online collected data. This approach, illustrated on Fig. 1, achieves the four desiderata: it stores light-weight data about past attempts, and uses an auxiliary model to update the action logits directly based on the previous outcomes. This allows MEMENTO to be used on top of any pretrained policy and to be totally agnostic to its architecture. In principle, MEMENTO can be combined with most existing construction RL algorithms: Attention-Model (Kool et al., 2019), POMO (Kwon et al., 2020), Poppy (Grinsztajn et al., 2023), COMPASS (Chalumeau et al., 2023b), which we confirm experimentally in Section 4.2. It learns an update rule during training, enabling data-efficient adaptation compared to policy gradient methods. The data retrieved from the memory and used by the auxiliary model to compute the new action logits contains more

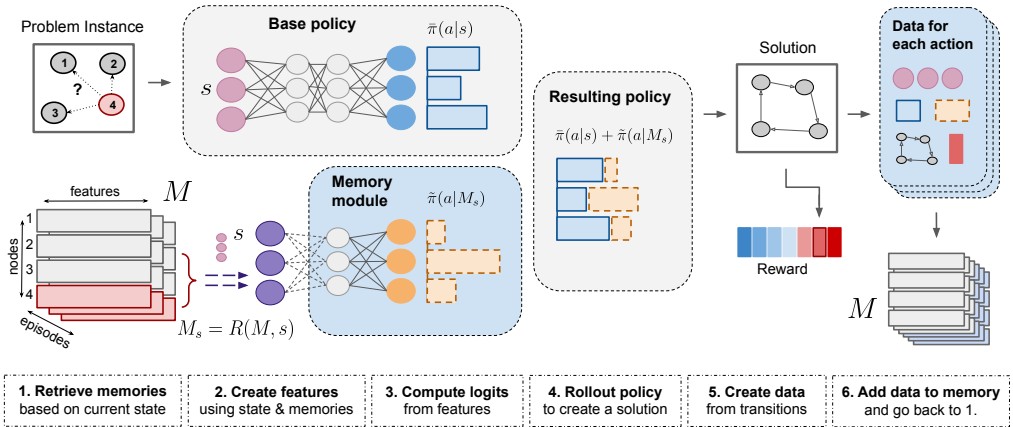

Figure 1: MEMENTO uses a memory to adapt neural solvers at inference time. When taking a decision, data from similar states is retrieved and prepared (1,2), then processed by a MLP to derive correction logits for each action (3). Summing the original and new logits enables to update the action distribution. The resulting policy is then rolled out (4), and transitions' data is stored in a memory (5,6), including node visited, action taken, log probability, and obtained return.

than the usual information used to derive a policy gradient update, for instance the budget remaining, enabling to calibrate the exploration/exploitation trade-off and discover superior update rules. We show in Appendix F that MEMENTO has capacity to at least rediscover REINFORCE, we provide a comparative plot on Fig. 4 and we show empirically in Section 4.1 that we outperform the leading policy-gradient updates adaptation method.

**Storing data in memory** In the memory, we store data about our past attempts. Akin to a replay buffer, this data needs to reflect past decisions, their outcome, and must enable to take better future decisions. In our memory, for each transition experienced while constructing a solution, we store the following: (i) node visited (ii) action taken, corresponding to the node that the policy decided to visit next (iii) log-probability given to that action by the model (iv) return of the entire trajectory (negative cost of the solution built) (v) budget at the time that solution was built (vi) log-probability that was suggested by the memory for that action (vii) log-probability associated with the entire trajectory and (viii) the log-probability associated with the remaining part of the trajectory. We can observe that elements (ii, iii and iv) are those needed to reproduce a REINFORCE update, and other features are additional context that will help for credit assignment and distribution shift estimation. Note that the remaining budget will also be added to the data when deriving the policy update.

**Retrieving data from the memory** Each time an action must be taken in a given state, we want to retrieve data in the memory that can inform us on past decisions that were taken in a similar situation. Since this retrieval process will happen at each step, we do not want it to be too costly. A good speed/relevance balance is achieved by retrieving data collected in the same node we are currently in; being faster than k-nearest neighbor retrieval, while extracting data of similar relevance.

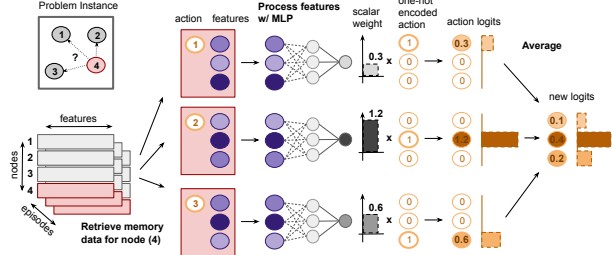

Figure 2: Building a new action distribution using the memory. Relevant data is retrieved and processed by a MLP to derive logits for each possible action.

**Processing data to update actions** Once data has been retrieved, it is used to update the action logits of the base policy. From the data, we separate the actions from their associated features (logp, return, etc...). We concatenate the remaining budget to the features. Each feature is normalised, and the resulting feature vector is processed by a Multilayer Perceptron (MLP) $H_{\theta_M}$ which outputs

a scalar weight. Each action is one-hot encoded, and a weighted average of the actions vectors is computed based on the scalar weight obtained. This aggregation outputs a new vector of action logits. This vector of logits is summed with the vector of logits output by the base model. The following paragraph introduces the mathematical formalism, and Fig. 2 illustrates it.

More formally, when visiting a node, with a given state $s$, we retrieve from the memory data experienced in the past when visiting that same node. This data, $M_s$, is a sequence of tuples $(a_i, f_{a_i})$, where $a_i$ are the past actions tried and $f_{a_i}$ are various features associated with the corresponding trajectories, as discussed above. The update is computed by $l_M = \sum_i \mathbf{a_i} H_{\theta_M}(f_{a_i})$, where $\mathbf{a_i}$ is the one-hot encoding of action $a_i$. Let $l$ be the logits from the base policy, the final logits used to sample the next action are given by $l + l_M$. We show in Appendix F that in the worst case, this enables MEMENTO to re-discover the REINFORCE update.

**Training** Existing construction methods are trained for one-shot optimization, with a few exceptions that are trained for few-shots (Grinsztajn et al., 2023; Chalumeau et al., 2023b; Hottung et al., 2024). We adapt the training process to take into account the budgeted multi-shot setting in which methods are actually used in practice. We hence rollout the policy as many times as the budget allows on the same instance before taking an update. Our loss is inspired by ECO-DQN (Barrett et al., 2020) and ECORD (Barrett et al., 2022): for a new given trajectory, the reward is the ReLU of the difference between the return and the best return achieved so far on that instance. Hence, those summed rewards equal the best score found over the budget. In practice, to account for the fact that it is harder to get an improvement as we get closer to the end of the budget, we multiply this loss with a weight that increases logarithmically as the budget is consumed. See Appendix B for explicit formula and Appendix B for pseudo-code.

**Inference** At the end of the training process, we obtain the parameters of the memory processing module, which can be used with any neural solver with no back-propagation needed anymore. The parameters of the base neural solver and the weights of the memory processing module are fixed, and the adaptation rules while simply be derived by filling the memory with the collected attempts and by processing the data retrieved from the memory. All hyperparameters can be found in Appendix C.

## 4 EXPERIMENTS

We evaluate our method across widely recognized CO problems Travelling Salesman (TSP) and Capacitated Vehicle Routing (CVRP). These problems serve as standard benchmarks for evaluating RL-based CO methods (Deudon et al., 2018; Kool et al., 2019; Kwon et al., 2020; Grinsztajn et al., 2023; Hottung et al., 2022; Choo et al., 2022; Chalumeau et al., 2023b). In Section 4.1, we validate our method by comparing it to leading approach for single-policy adaptation using online collected data, EAS (Hottung et al., 2022). We present their comparative performance in line with established standard benchmarking from literature, and elucidate the mechanisms employed to adapt the action distribution of the base policy during inference. We then demonstrate how MEMENTO can be combined with SOTA neural solver COMPASS with no additional retraining in Section 4.2. Finally, we scale those methods to larger instances in Section 4.3, outperforming heatmap-based methods.

**Code availability** We provide access to the code[1] utilized for training our method and executing all baseline models. Additionally, we release our checkpoints for all problem types and scales, accompanied by the necessary datasets to replicate our findings. We implement our method and experiments in JAX (Bradbury et al., 2018), in a codebase that is compatible with previous JAX implementation released in the community. The two problems are also JAX implementations from Jumanji (Bonnet et al., 2023), enabling us to fully leverage hardware accelerators such as TPUs. Our implementations are optimized for TPU v3-8, aligning with the hardware utilized in our experiments.

### 4.1 BENCHMARKING MEMENTO AGAINST POLICY GRADIENT FINE-TUNING

The most direct way to leverage online data for policy update is RL/policy-gradients. By contrast, MEMENTO learns an update based on the collected data. We therefore want to see how those two

---

[1]Code, checkpoints, and evaluation sets available at *anonymised for the review process*

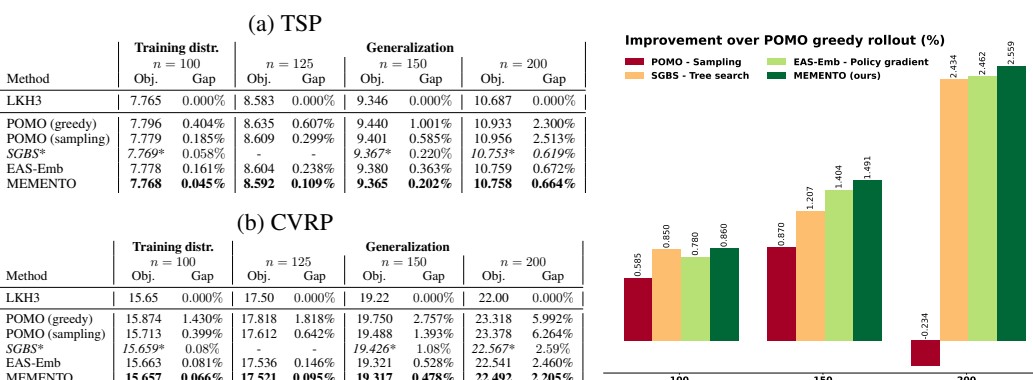

**(a) TSP**

| Method | Training distr. $n=100$ | | Generalization $n=125$ | | $n=150$ | | $n=200$ | |
|---|---|---|---|---|---|---|---|---|
| | Obj. | Gap | Obj. | Gap | Obj. | Gap | Obj. | Gap |
| LKH3 | 7.765 | 0.000% | 8.583 | 0.000% | 9.346 | 0.000% | 10.687 | 0.000% |
| POMO (greedy) | 7.796 | 0.404% | 8.635 | 0.607% | 9.440 | 1.001% | 10.933 | 2.300% |
| POMO (sampling) | 7.779 | 0.185% | 8.609 | 0.299% | 9.401 | 0.585% | 10.956 | 2.513% |
| *SGBS** | *7.769** | *0.058%* | - | - | *9.367** | *0.220%* | *10.753** | *0.619%* |
| EAS-Emb | 7.778 | 0.161% | 8.604 | 0.238% | 9.380 | 0.363% | 10.759 | 0.672% |
| MEMENTO | **7.768** | **0.045%** | **8.592** | **0.109%** | **9.365** | **0.202%** | **10.758** | **0.664%** |

**(b) CVRP**

| Method | Training distr. $n=100$ | | Generalization $n=125$ | | $n=150$ | | $n=200$ | |
|---|---|---|---|---|---|---|---|---|
| | Obj. | Gap | Obj. | Gap | Obj. | Gap | Obj. | Gap |
| LKH3 | 15.65 | 0.000% | 17.50 | 0.000% | 19.22 | 0.000% | 22.00 | 0.000% |
| POMO (greedy) | 15.874 | 1.430% | 17.818 | 1.818% | 19.750 | 2.757% | 23.318 | 5.992% |
| POMO (sampling) | 15.713 | 0.399% | 17.612 | 0.642% | 19.488 | 1.393% | 23.378 | 6.264% |
| *SGBS** | *15.659** | *0.08%* | - | - | *19.426** | *1.08%* | *22.567** | *2.59%* |
| EAS-Emb | 15.663 | 0.081% | 17.536 | 0.146% | 19.321 | 0.528% | 22.541 | 2.460% |
| MEMENTO | **15.657** | **0.066%** | **17.521** | **0.095%** | **19.317** | **0.478%** | **22.492** | **2.205%** |

Figure 3: Results of MEMENTO against baselines on a standard benchmark comprising 8 datasets with 4 distinct instance sizes on TSP (a) and CVRP (b). Methods are evaluated on instances from training distribution (100) as well as on larger instance sizes to test generalization. The right panel aggregates those results, reported in % improvement relative to a greedy rollout of the given base policy (POMO). MEMENTO is consistently outperforming policy-gradient based method EAS and tree-search based method SGBS, with significant improvement over the base policy. *SGBS results are reported from Choo et al. (2022), inducing biases in the comparison, discussed in Appendix A.1.

approaches compare. Since EAS (Hottung et al., 2022) is SOTA method using policy gradient, we benchmark MEMENTO against it on a standard set of instances used in the literature (Kool et al., 2019; Kwon et al., 2020; Hottung et al., 2022). Specifically, for TSP and CVRP, those datasets comprise 10 000 instances drawn from the training distribution. These instances feature the positions of 100 cities/customers uniformly sampled within the unit square. The benchmark also includes three datasets of distributions not encountered during training, each comprising of 1000 problem instances with larger sizes: 125, 150, and 200, generated from a uniform distribution across the unit square. We employ the exact same datasets as those utilized in the literature for consistency and comparability.

**Setup** For routing problems like TSP and CVRP, POMO (Kwon et al., 2020) is the base single-agent, one-shot architecture that underpins most RL construction solvers. In this set of experiments, MEMENTO and EAS both use POMO as a base policy, and adapt it to get the best possible performance within a given budget. We train MEMENTO until it converges, on the same instance distribution as that used for the initial checkpoint. When assessing active-search performance, each method operates within a fixed budget of 1600 attempts, a methodology akin to Hottung et al. (2022). In this setup, each attempt comprises of one trajectory per possible starting point. This standardized approach facilitates direct comparison to POMO and EAS, which also utilize rollouts from all starting points at each step. Following recent works, we do not use augmentations with symmetries which have proved to not be critical and greatly increase the cost (Chalumeau et al., 2023b).

**Results** The average performance of each method across all problem settings are presented in Fig. 3. The observations we draw are three-fold. First, MEMENTO outperforms its base model (POMO) on the entire benchmark by a significant margin: showing that its adaptive search is superior to stochastic sampling. Second, MEMENTO outperforms EAS on all 8 tasks (spanning both in- and out-of-distribution) for both environments, highlighting the efficacy of learned policy updates when compared to vanilla policy gradients. Finally, we note that this improvement is significant; for example, on TSP100, MEMENTO is doing 60% better than sampling, while EAS only 6%.

**Analysing the update rule** To better understand how MEMENTO uses its memory to adapt the action distribution of its base policy, we analyse the logit update that an action gets with respect to its associated data, akin to similar analysis conducted in meta-learning (Lu et al., 2022; Jackson et al., 2024), and compare it to the REINFORCE policy gradient. On Fig. 4, we plot the heatmap of the logit update with respect to the log probability (logp) and return associated with an action. We observe that the main rules learned by MEMENTO are similar to REINFORCE: an action with low logp and high return gets a positive update while an action with high logp and low return gets

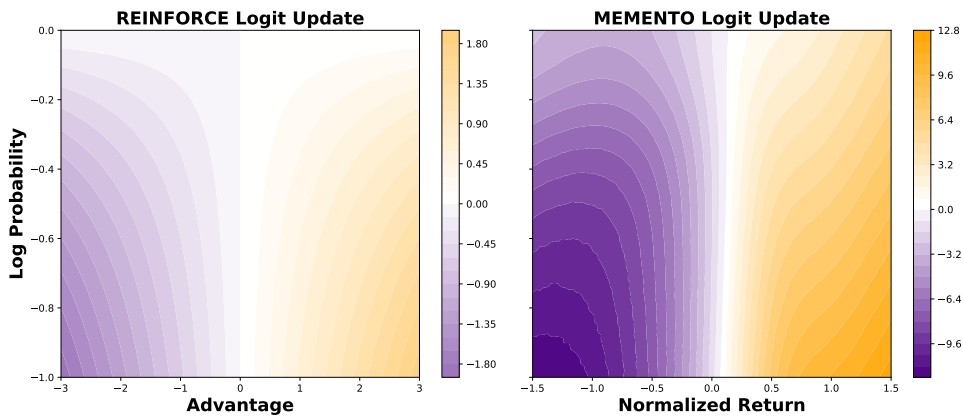

Figure 4: Heatmaps of the update rules induced by REINFORCE gradient (left), compared to those learned by MEMENTO (right). Akin to REINFORCE, MEMENTO encourages actions with high returns, particularly when they have low probability. Interestingly, MEMENTO learns an asymmetric rule: requiring the normalised return to be strictly positive to reinforce an action, but encouraging it even more when that condition is reached (higher amplitude for positive updates than negative ones).

discouraged. It is nonetheless interesting to see discrepancies. In particular, MEMENTO's rules are not symmetric with respect to $x = 0$, it only encourages action that are above the mean return, but gives higher amplitude to high-return actions than it discourages low-return ones, pushing for performance while preserving exploration. A similar dissymetry would be expected with EAS since it combines REINFORCE with a term that increases the likelihood to generate the best solution found.

Additionally, MEMENTO uses more inputs than REINFORCE: current budget available, trajectory logp, age of the data, and previous MEMENTO logit are also used to derive the new action logit. Appendix G provides an ablation study of those additional inputs, validating their importance.

### 4.2 ZERO-SHOT COMBINATION OF MEMENTO WITH UNSEEN SOLVERS

MEMENTO was designed to be agnostic to the base model, enabling combination with existing and future solvers. In this section, we demonstrate zero-shot combination (no re-training) with state-of-the-art solver COMPASS. COMPASS provides fast adaptation since it uses a collection of diverse pre-trained policies that can be searched to pick the most relevant policy for unseen instances.

Although providing fast adaptation, it can hardly improve further once the appropriate pre-trained policy has been found. In particular, online collected data cannot be used to update that particular policy's action distribution. We use MEMENTO's memory module, trained with POMO as a base policy, and apply it on COMPASS with no additional re-training, and show empirically that we can switch on MEMENTO's adaptation mechanism during the search and reach a new state-of-the-art on 11 out of 12 tasks.

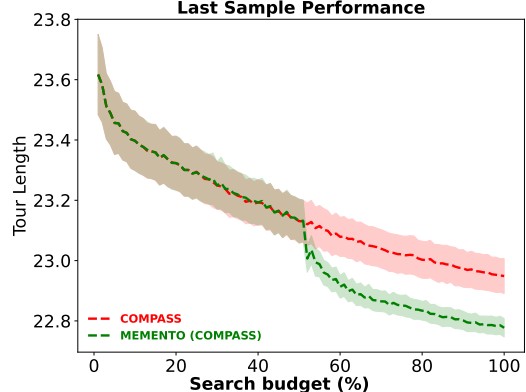

Figure 5: Evolution of performance over budget when combining MEMENTO to COMPASS on CVRP200, no re-training needed.

**Methodology** We combine our checkpoint of MEMENTO with a released model of COMPASS (Chalumeau et al., 2023b). For the first half of the budget, we do not use MEMENTO, to let COMPASS find the appropriate pre-trained policy. Once the search starts narrowing down, we activate MEMENTO to adapt the pre-trained policy, with no need to turn off COMPASS' search. Fig. 5 shows the typical trends

observed at inference. We see the typical search reported in Chalumeau et al. (2023b). At 50% of budget consumption, MEMENTO is activated and one can observe a clear and significant drop in averaged tour length of the latest sampled solutions. The standard deviation also illustrates how the search is suddenly narrowed down.

Table 1: Zero-shot combination of MEMENTO and COMPASS on CVRP. Rules learned by ME-MENTO transfer to population-based method COMPASS and achieve SOTA on standard benchmark.

| Method | $n = 100$ | | $n = 125$ | | $n = 150$ | | $n = 200$ | |
| | Obj. | Gap | Obj. | Gap | Obj. | Gap | Obj. | Gap |
|---|---|---|---|---|---|---|---|---|
| COMPASS | 15.644 | -0.019% | 17.511 | 0.064% | 19.313 | 0.485% | 22.462 | 2.098% |
| MEMENTO (COMPASS) | **15.634** | **-0.082%** | **17.497** | **-0.041%** | **19.290** | **0.336%** | **22.405** | **1.808%** |

**Results**  We evaluate the zero-shot combination of MEMENTO and COMPASS on the standard benchmark reported in Section 4.1. Results for CVRP are presented in Table 1, showing tour length and optimality gap in-distribution (CVRP100) and out-of-distribution. Although requiring no re-training, MEMENTO's rules combine efficiently with COMPASS and achieves new SOTA. Results on TSP are reported in Appendix A, and results on larger instances (TSP & CVRP) in Section 4.3.

## 4.3 SCALING RL CONSTRUCTION METHODS TO LARGER INSTANCES

Scaling neural solvers to larger instances is a crucial challenge to be tackled in order to bridge the gap to industrial applications. Auto-regressive construction-based solvers are promising approaches, requiring limited expert knowledge whilst providing strong performance. But are still hardly ever trained on larger scales with RL. As a result, their performance reported in the literature is usually quite unfair, being evaluated in large scales instances (>500) although having been trained on 100 nodes. In this section, we explain how we have trained construction solvers POMO and COMPASS on instances of size 500 with RL (for both TSP and CVRP, checkpoints to be open-sourced); and then use them as new base models to train and validate properties of MEMENTO at that scale.

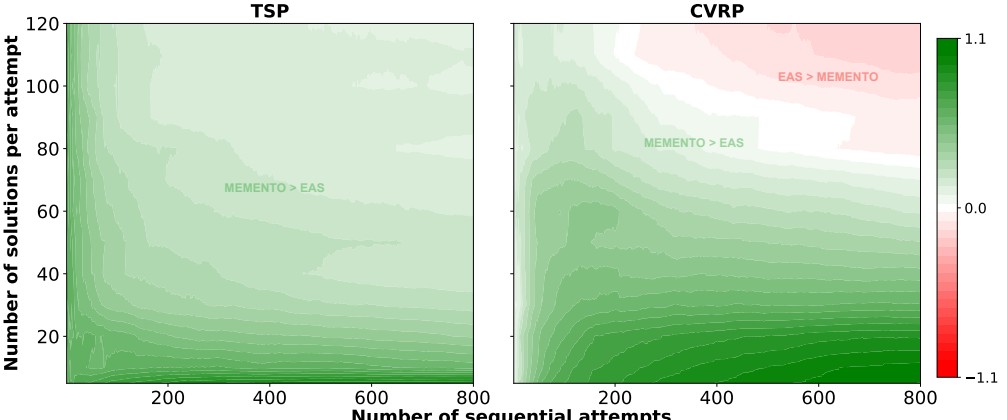

Figure 6: Performance improvement of MEMENTO over EAS on instances of size 500 on TSP (left) and CVRP (right) for increasing number of sequential attempts and sizes of attempt batch. Green areas correspond to settings for which MEMENTO's online adaptation is more efficient than EAS. MEMENTO always outperforms EAS on TSP, and for most settings on CVRP. The difference is particularly significant for low budgets, showing the data-efficiency and robustness of MEMENTO.

**RL training on larger instances**  Training POMO on instances of size 500 involves three main steps. First, inspired by curriculum-based methods, we start from a checkpoint pre-trained on TSP100. Second, we use Efficient Attention (Rabe & Staats, 2022) to reduce the memory cost of multi-head attention, enabling to rollout problems in parallel despite the $O(n^2)$ memory requirement. Third, we use gradient accumulation to keep good estimates despite the constrained smaller instance batch size. These three combined tricks enable to train POMO till convergence within 4 days, and consequently

to train COMPASS and MEMENTO. We also build the zero-shot combined MEMENTO (COMPASS) checkpoint. We release all checkpoints (TSP & CVRP) publicly to ease future research at this scale.

**Inference-time constraints** On those larger instances, the cost of constructing a solution is significantly increased (for POMO's architecture: squared in computation and memory, and linear in sequential operations). It is also harder to get numerous parallel shots, which most methods depend on. It hence becomes crucial to develop data-efficient methods which stay robust to low budget regimes; and important to know which method to use for each budget constraint.

**MEMENTO on larger instances** We compare MEMENTO and EAS on a set of 128 unseen instances (of size 500), for a range of budget expressed in number of sequential attempts and number of parallel attempts attempts. Figure 6 reports the percentage of improvement (in absolute cost) brought by MEMENTO over EAS. From this figure, we can observe three main tendencies. First, the data-efficiency of MEMENTO is illustrated by its consistent superiority on low budget (lower left of plots), reaching significant improvement compared to EAS. Then, we can see that, on CVRP, the gap increases with the sequential budget for low parallelism. It demonstrates that MEMENTO's update stays robust, whereas the gradient estimates of EAS get deteriorated and cannot bring further improvement despite additional sequential budget. To finish with, we can see that MEMENTO fully dominates the heatmap on TSP, even for higher budget. Nevertheless, when budget for CVRP becomes bigger, EAS is able to outperform MEMENTO, given that batches are large enough ($> 80$).

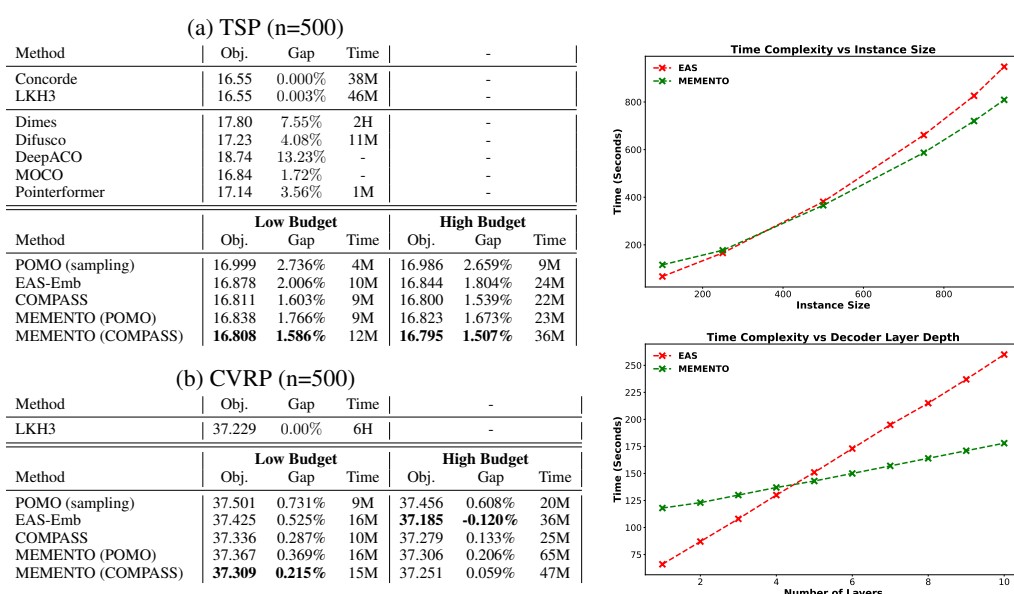

(a) TSP (n=500)

| Method | Obj. | Gap | Time | - |
|---|---|---|---|---|
| Concorde | 16.55 | 0.000% | 38M | - |
| LKH3 | 16.55 | 0.003% | 46M | - |
| Dimes | 17.80 | 7.55% | 2H | - |
| Difusco | 17.23 | 4.08% | 11M | - |
| DeepACO | 18.74 | 13.23% | - | - |
| MOCO | 16.84 | 1.72% | - | - |
| Pointerformer | 17.14 | 3.56% | 1M | - |

| Method | Low Budget | | | High Budget | | |
|---|---|---|---|---|---|---|
| | Obj. | Gap | Time | Obj. | Gap | Time |
| POMO (sampling) | 16.999 | 2.736% | 4M | 16.986 | 2.659% | 9M |
| EAS-Emb | 16.878 | 2.006% | 10M | 16.844 | 1.804% | 24M |
| COMPASS | 16.811 | 1.603% | 9M | 16.800 | 1.539% | 22M |
| MEMENTO (POMO) | 16.838 | 1.766% | 9M | 16.823 | 1.673% | 23M |
| MEMENTO (COMPASS) | **16.808** | **1.586%** | 12M | **16.795** | **1.507%** | 36M |

(b) CVRP (n=500)

| Method | Obj. | Gap | Time | - |
|---|---|---|---|---|
| LKH3 | 37.229 | 0.00% | 6H | - |

| Method | Low Budget | | | High Budget | | |
|---|---|---|---|---|---|---|
| | Obj. | Gap | Time | Obj. | Gap | Time |
| POMO (sampling) | 37.501 | 0.731% | 9M | 37.456 | 0.608% | 20M |
| EAS-Emb | 37.425 | 0.525% | 16M | **37.185** | **-0.120%** | 36M |
| COMPASS | 37.336 | 0.287% | 10M | 37.279 | 0.133% | 25M |
| MEMENTO (POMO) | 37.367 | 0.369% | 16M | 37.306 | 0.206% | 65M |
| MEMENTO (COMPASS) | **37.309** | **0.215%** | 15M | 37.251 | 0.059% | 47M |

Figure 7: Tables on the left report results of MEMENTO against the baseline algorithms on instances of size 500 of TSP (a) and CVRP (b) for two budget regimes (low and high). We observe that POMO-based methods are outperforming concurrent methods. MEMENTO achieves single-agent SOTA (resp. overall SOTA) on 3 out of 4 settings when combined with POMO (resp. COMPASS). The plots on the right show the time complexity of MEMENTO and EAS for increasing values of decoder size, and instance size. It demonstrates that MEMENTO scales better in time than EAS.

**Results** In Fig. 7, we report performance of our checkpoints, using datasets introduced in Fu et al. (2021) and commonly used in the literature. For each environment, we report the results on two different setting: (i) Low Budget, where the methods are given $25\,000$ attempts, (ii) High Budget, where $100\,000$ attempts are available. We also report results from concurrent RL methods (Qiu et al., 2022; Dernedde et al., 2024; Sun & Yang, 2023; Ye et al., 2023), without method-agnostic local search; and industrial solvers LKH3 (Helsgaun, 2017) and Concorde (Applegate et al., 2006).

First, we observe that stochastically sampling solutions with POMO for less than 10 minutes already provides competitive results, ranking second among the five leading neural solvers. Adding EAS on top of POMO enables to compete with leading method MOCO on TSP500. This shows that auto-regressive RL constructive methods are already competitive at this scale. It is the first time that POMO is trained on those instance sizes, making it the first RL-trained neural solver that can be used for large instances both on TSP and CVRP, since most previous large scale RL methods are graph-specific, and hence cannot be applied on CVRP.

Furthermore, those results confirm that MEMENTO's adaptation performance scales to instance size, and that it can still be zero-shot combined with COMPASS. On this benchmark, MEMENTO achieved state-of-the-art on 3 of the 4 tasks when zero-shot combined with COMPASS, showing significant improvement compared to previous state-of-the-art MOCO: the gap to optimality is reduced by more than $10\%$. It also proves efficiency on those large scales by being the best single agent method on all TSP instances, and on the low budget CVRP task.

**Time complexity** Time performance at scale is key for the wide adoption of neural solvers since the number of sequential batches of attempts that can be achieved within a time budget will mostly depend on it (plus on the accessible hardware). Adaptation methods must be able to tackle large instances in reasonable time, and they must also scale well with the size of the base solver used, since Luo et al. (2023) demonstrated the importance of using large and multi-layered decoders for performance in neural CO (although they could not yet train them with RL).

Hence, we evaluate MEMENTO and EAS on a set of increasing instance sizes and decoder sizes and report their time complexity on the right part of Fig. 7. These plot show that, despite being slower for small settings, MEMENTO's adaptation mechanism scales better than EAS. For an instance size of 1000, MEMENTO becomes $20\%$ faster to use the budget. And even for small instances (size 100), using 10 layers in the decoder's architecture (1M parameters) makes EAS $40\%$ slower than MEMENTO. These scaling laws illustrate another benefit of using an approach that learns the update rather than relying on back-propagation to adapt neural solvers at inference time.

## 5 CONCLUSION

We present MEMENTO, a method to improve adaptation of neural CO solvers to unseen instances. MEMENTO enables to condition a policy directly on data collected online during search, facilitating rapid adaptation and efficient use of the search budget. In practice, it proves to outperform stochastic sampling, tree search and policy-gradient fine-tuning of the policy, and shows it can combined with another solver than the one used during training, with no further re-training needed. Additionally, MEMENTO respects several key properties, like robustness to low-budget regimes, and favorable time performance scalability, both in instance size and base solver size.

We demonstrate the efficacy of MEMENTO by achieving state-of-the-art single-policy adaptation on a standard benchmark on TSP and CVRP, both in and out-of-distribution. Moreover, we find that MEMENTO realises interpretable update rules to the underlying policy; trading off exploration and exploitation over the search budget and outperforming REINFORCE-style updates. We further demonstrate the flexibility of this approach by achieving zero-shot combination of MEMENTO and COMPASS, achieving overall state-of-the-art on a benchmark of 12 tasks. The simplicity and efficacy of this method makes it amenable to a very large set of problems, laying another stone towards the application of neural solvers to real-world problems.

**Limitations and future work.** MEMENTO incurs additional compute and memory-usage compared to the memory-less base policies which it augments. Practically, we find that the performance gain significantly outweighs these overheads. A potential mitigation could be to derive the mathematical update learned by MEMENTO to avoid relying on the MLP computation. The adaptation enabled by MEMENTO is determined by the data collected by the base policy, therefore this could limit the efficiency when evaluation instances are very far from the training distribution, or when the base policy has harmfully poor initial performance. Finally, we believe that the capacity of MEMENTO could be improved by training on broader distributions, in particular using variable instance sizes.

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

APPENDIX

# A  EXTENDED RESULTS

In Section 4, we compare MEMENTO to popular methods from the literature in numerous settings. In particular, Figure 3 compares using stochastic sampling, MEMENTO and EAS to adapt POMO on the standard benchmark; and Figure 6 compares the relative performances of MEMENTO and EAS on larger instances of TSP and CVRP for several values of batch sizes and budget. In this section, we report additional values and results of other methods for those experiments.

## A.1  STANDARD BENCHMARK

We evaluate our method, MEMENTO, against leading RL methods and industrial solvers. For RL specific methods, we provide results for POMO with greedy action selection and stochastic sampling, and EAS; an active search RL method built on top of POMO, that fine-tunes a policy on each problem instance. We also report population-based method COMPASS and the zero-shot combination of COMPASS with MEMENTO.

We compare to the heuristic solver LKH3 (Helsgaun, 2017); the current leading industrial solver of both TSP and CVRP, as well as an exact solver Concorde (Applegate et al., 2006) which is a TSP-specific industrial solver, and the CVRP-specific solver (Vidal et al., 2012).

We use datasets of 10,000 instances with 100 cities/customer nodes drawn from the training distribution, and three generalization datasets of 1,000 instances of sizes 125, 150, and 200, all from benchmark sets frequently used in the literature (Kool et al., 2019; Kwon et al., 2020; Hottung et al., 2022; Grinsztajn et al., 2023; Chalumeau et al., 2023b). Tables 2a and 2b display results for TSP and CVRP on the standard benchmark, respectively. The columns provide the absolute tour length, the optimality gap, and the total inference time that each method takes to solve one instance within the attempts budget.

**Times reported**  When possible, we decide to report the time to solve one instance rather the entire dataset following four main observations: (i) First, the literature use datasets of varying sizes, e.g. 10k for TSP100, 1k for TSP200, 128 for TSP500, hence time reported can be confusing, and do not enable to clearly see how methods' solving time scale with instance size. (ii) Second, the default real-world application consist in solving one instance at a time, or solving several instances at the same time but on separated hardware. (iii) Additionally, measuring on the entire dataset mixes several aspects, i.e. the instance solving time is mixed with the batch scalability of the method. We do think that this property is very interesting to know, but should be considered on the side, rather than mixed with the instance solving time. (iv) Moreover, this property will express differently depending on the hardware available. For instance, on a small hardware POMO sampling with $n$ attempts will be $n$ times slower than POMO greedy; but given a large enough GPU, POMO sampling can be parallelised $n$ times and hence take exactly the same amount of time as POMO greedy.

We were able to do so on the standard benchmark since we have implementations of most methods we were comparing in a single framework. Since we are reporting several external methods in Section 4.3, we could only report time taken for the full dataset in that case.

**Additional comments about the results**  On Table 2, we can see that in the single-agent setting, MEMENTO leads the entire benchmark, and in the population-based setting, MEMENTO (COMPASS) is leading the whole benchmark. MEMENTO is able to give significant improvement to COMPASS on CVRP, pushing significantly the state-of-the-art on this benchmark. On TSP, the improvement is not significant enough to be visible on the rounded results. To finish with, we can observe that the time cost associated with MEMENTO is reasonable and is worth the performance improvement (except maybe for TSP results of MEMENTO (COMPASS)).

**Notes concerning SGBS**  All neural methods reported in Table 2 are our own runs with standardised checkpoints and rollout strategies, except for the results of SGBS, which were taken from Choo et al. (2022). This introduces three biases: (i) the base POMO checkpoint used by SGBS is not exactly the

same as our re-trained POMO checkpoint (ii) SGBS uses the domain-specific augmentation trick that we do not use (iii) SGBS pre-selects starting points in CVRP, which we do not do.

Overall, the results reported in SGBS show that SGBS alone is always outperformed by EAS; hence, it should be expected that in all the settings where we outperform EAS, we would significantly outperform SGBS if both methods were evaluated exactly in the same way. We hence expect the gap between MEMENTO and SGBS to be larger than the one reported here. Additionally, SGBS has yet never been validated on larger instances. Nevertheless, we think that SGBS is a very efficient method from the NCO toolbox and appreciate that MEMENTO and SGBS are orthogonal, and could be combined for further improvement. We leave this for future work.

Table 2: Results of MEMENTO against the baseline algorithms for (a) TSP, (b) CVRP. The methods are evaluated on instances from training distribution ($n = 100$) as well as on larger instance sizes to test generalization. We use the same dataset as the rest of the literature, those contain $10\,000$ instances for $n = 100$ and 1000 instances for $n = 125, 150, 200$. Gaps are computed relative to LKH3. We report time needed to solve one instance. *SGBS results are reported from Choo et al. (2022), and do not use the same POMO checkpoint as other reported results. Additionally, they rely on problem-specific tricks that were not used by other methods. Details in Appendix A.1.

(a) TSP

| | Training distr. | | | Generalization | | | | | | | | |
| | $n = 100$ | | | $n = 125$ | | | $n = 150$ | | | $n = 200$ | | |
| Method | Obj. | Gap | Time | Obj. | Gap | Time | Obj. | Gap | Time | Obj. | Gap | Time |
|---|---|---|---|---|---|---|---|---|---|---|---|---|
| Concorde | 7.765 | 0.000% | 0.49S | 8.583 | 0.000% | 0.72S | 9.346 | 0.000% | 1S | 10.687 | 0.000% | 1.86S |
| LKH3 | 7.765 | 0.000% | 2.9S | 8.583 | 0.000% | 4.4S | 9.346 | 0.000% | 6S | 10.687 | 0.000% | 11S |
| POMO (greedy) | 7.796 | 0.404% | 0.16S | 8.635 | 0.607% | 0.2S | 9.440 | 1.001% | 0.29S | 10.933 | 2.300% | 0.45S |
| POMO (sampling) | 7.779 | 0.185% | 16S | 8.609 | 0.299% | 20S | 9.401 | 0.585% | 29S | 10.956 | 2.513% | 45S |
| *SGBS** | *7.769** | *0.058%** | - | - | - | - | *9.367** | *0.220%** | - | *10.753** | *0.619%** | - |
| EAS | 7.778 | 0.161% | 39S | 8.604 | 0.238% | 46S | 9.380 | 0.363% | 64S | 10.759 | 0.672% | 91S |
| MEMENTO (POMO) | 7.768 | 0.045% | 43S | 8.592 | 0.109% | 52S | 9.365 | 0.202% | 77S | 10.758 | 0.664% | 115S |
| COMPASS | 7.765 | 0.008 % | 20S | 8.586 | 0.036% | 24S | 9.354 | 0.078% | 33S | 10.724 | 0.349% | 49S |
| MEMENTO (COMPASS) | **7.765** | **0.008%** | 45S | **8.586** | **0.035%** | 55S | **9.354** | **0.077%** | 83S | **10.724** | **0.348%** | 127S |

(b) CVRP

| | Training distr. | | | Generalization | | | | | | | | |
| | $n = 100$ | | | $n = 125$ | | | $n = 150$ | | | $n = 200$ | | |
| Method | Obj. | Gap | Time | Obj. | Gap | Time | Obj. | Gap | Time | Obj. | Gap | Time |
|---|---|---|---|---|---|---|---|---|---|---|---|---|
| HGS | 15.563 | $-0.536\%$ | 19S | - | - | - | 19.055 | $-0.884\%$ | 32S | 21.766 | $-1.096\%$ | 61S |
| LKH3 | 15.646 | 0.000% | 52S | 17.50 | 0.000% | - | 19.222 | 0.000% | 72S | 22.003 | 0.000% | 90S |
| POMO (greedy) | 15.874 | 1.430% | 24S | 17.818 | 1.818% | 34S | 19.750 | 2.757% | 52S | 23.318 | 5.992% | 87S |
| POMO (sampling) | 15.713 | 0.399% | 24S | 17.612 | 0.642% | 34S | 19.488 | 1.393% | 52S | 23.378 | 6.264% | 87S |
| *SGBS** | *15.659** | *0.08%** | - | - | - | - | *19.426** | *1.08%** | - | *22.567** | *2.59%** | - |
| EAS | 15.663 | 0.081% | 66S | 17.536 | 0.146% | 82S | 19.321 | 0.528% | 123S | 22.541 | 2.460% | 179S |
| MEMENTO (POMO) | 15.657 | 0.066% | 118S | 17.521 | 0.095% | 150S | 19.317 | 0.478% | 169S | 22.492 | 2.205% | 392S |
| COMPASS | 15.644 | -0.019% | 29S | 17.511 | 0.064% | 39S | 19.313 | 0.485% | 56S | 22.462 | 2.098% | 85S |
| MEMENTO (COMPASS) | **15.634** | **-0.082%** | 136S | **17.497** | **-0.041%** | 162S | **19.290** | **0.336%** | 180S | **22.405** | **1.808%** | 460S |

## A.2 PERFORMANCE OVER DIFFERENT NUMBER OF PARALLEL AND SEQUENTIAL ATTEMPTS

In Section 4.3, we show performance improvement of MEMENTO over EAS on instances of size 500 on TSP and CVRP for increasing number of sequential attempts and size of attempt batch. We report extended results with an additional competitor, POMO. We compare the online adaptation of the methods over four different sizes of batched solutions across increasing sequential attempts for each CO problem. Results from Tables 3a and 3b show results of the three methods on TSP and CVRP, respectively. The columns show the best tour length performance for various values of sequential attempts expressed as budget, and solution batches of size $N$. Performance is averaged over a set of 128 instances.

## A.3 EVALUATION OVER LARGER INSTANCES

In Section 4.3, we evaluate MEMENTO and baselines on instances of size 500. For TSP, we use the dataset from Fu et al. (2021). For CVRP, we use the dataset from Luo et al. (2023). We do not include LEHD in our results since we focus on methods trained with Reinforcement Learning, and LEHD can only be successfully trained with supervised learning at the time of writing. Nevertheless,

Table 3: Results of MEMENTO, POMO and EAS on instances of size 500 of (a) TSP and (b) CVRP for increasing number of sequential attempts and sizes of attempt batch.

(a) TSP

| Method | $N = 20$ | | | | | $N = 40$ | | | | |
|---|---|---|---|---|---|---|---|---|---|---|
| | 200 | 400 | 600 | 800 | 1000 | 200 | 400 | 600 | 800 | 1000 |
| POMO (sampling) | 16.9810 | 16.9707 | 16.9638 | 16.9619 | 16.9574 | 16.9717 | 16.96 | 16.9549 | 16.9523 | 16.9493 |
| EAS | 16.9223 | 16.8923 | 16.8754 | 16.864 | 16.8556 | 16.8777 | 16.8468 | 16.83 | 16.8208 | 16.8143 |
| MEMENTO | **16.8312** | **16.81** | **16.799** | **16.7939** | **16.7902** | **16.8187** | **16.8018** | **16.7926** | **16.7855** | **16.7815** |

| Method | $N = 60$ | | | | | $N = 80$ | | | | |
|---|---|---|---|---|---|---|---|---|---|---|
| | 200 | 400 | 600 | 800 | 1000 | 200 | 400 | 600 | 800 | 1000 |
| POMO (sampling) | 16.9678 | 16.9587 | 16.9539 | 16.952 | 16.9503 | 16.9634 | 16.9547 | 16.9513 | 16.9495 | 16.9474 |
| EAS | 16.8616 | 16.8353 | 16.8211 | 16.8134 | 16.8084 | 16.8521 | 16.8234 | 16.8106 | 16.8014 | 16.7941 |
| MEMENTO | **16.8129** | **16.7966** | **16.7867** | **16.7816** | **16.7780** | **16.8087** | **16.7935** | **16.7854** | **16.7811** | **16.7770** |

(b) CVRP

| Method | $N = 20$ | | | | | $N = 40$ | | | | |
|---|---|---|---|---|---|---|---|---|---|---|
| | 200 | 400 | 600 | 800 | 1000 | 200 | 400 | 600 | 800 | 1000 |
| POMO (sampling) | 37.5256 | 37.4917 | 37.4763 | 37.4639 | 37.4570 | 37.4858 | 37.4599 | 37.4475 | 37.4372 | 37.4307 |
| EAS | 37.6107 | 37.5988 | 37.5914 | 37.582 | 37.5769 | 37.5022 | 37.4546 | 37.422 | 37.4017 | 37.3849 |
| MEMENTO | **37.3398** | **37.2992** | **37.2731** | **37.2589** | **37.2527** | **37.3172** | **37.2776** | **37.2515** | **37.2357** | **37.2260** |

| Method | $N = 60$ | | | | | $N = 80$ | | | | |
|---|---|---|---|---|---|---|---|---|---|---|
| | 200 | 400 | 600 | 800 | 1000 | 200 | 400 | 600 | 800 | 1000 |
| POMO (sampling) | 37.4598 | 37.436 | 37.4228 | 37.4191 | 37.4135 | 37.4588 | 37.4335 | 37.4198 | 37.4113 | 37.4050 |
| EAS | 37.4569 | 37.3679 | 37.3248 | 37.2892 | 37.2624 | 37.3588 | 37.2804 | **37.2346** | **37.2021** | **37.1747** |
| MEMENTO | **37.292** | **37.2607** | **37.2305** | **37.2171** | **37.2092** | **37.2887** | **37.2556** | 37.2382 | 37.2268 | 37.2142 |

the good performance achieved by LEHD has motivated us to study the scaling law of MEMENTO as the number of layers in the decoder increases, reported on Section 4.3.

In order to run COMPASS and MEMENTO (COMPASS) with the same batch sizes as other methods on those large instances, we reduced the number of starting points used by COMPASS to ensure that this number multiplied by the number of latent vector sampled at the same time is equal to the number of starting point used by other methods (POMO, EAS, MEMENTO). In practice, we used a latent vector batch of size 10, and hence divided the number of starting points by 10. This is slightly disadvantaging COMPASS and MEMENTO (COMPASS) but enables to respect the constraint of number of parallel batches that can be achieved at once. Note that this slightly impacts the time performance reported since JAX jitting process will not fuse the operations in the same way, additionally, when combined with MEMENTO, this impact the size of the memory (we keep one per starting point to adapt to POMO, although this is completely agnostic to MEMENTO's method in itself). Since those factors impacts TSP and CVRP performance in different ways, this explains why their relative speed differ, i.e. why MEMENTO (COMPASS) is slower on TSP but faster on CVRP.

### A.4 TIME COMPLEXITY ANALYSIS

To get the time curves reported in Section 4.3, we used CVRP, since it is the environment were MEMENTO was the slowest compared to EAS. We hence expect curves on TSP to be even better for MEMENTO. For the instance size scaling, we use 50 starting points and 100 sequential attempts, and 1 layer in the decoder, and evaluate several instance sizes going from 100 to 1000. For the decoder layer depth scaling, we use CVRP100, 100 starting points and 160 sequential attempts. We then evaluate methods for a depth going from 1 to 10.

## B TRAINING PROCEDURE

This section give a detailed description of how an existing pre-trained model can be augmented with MEMENTO, and how we train MEMENTO to acquire adaptation capacities.

Firstly, we take an existing single-agent model that was trained using the REINFORCE algorithm and reuse it as a base model. In our case, the base model is POMO. We augment POMO with a memory module and begin the training procedure which aims to create a policy that is able to use past experiences to make decisions in a multi-shot setting. We initialize the memory module weights with

small values such that they barely affect the initial output. Hence, the initial MEMENTO checkpoint maintains the same performance as the pre-trained POMO checkpoint.

In the training procedure, the policy is trained to use data stored in the memory to take decisions and solve a problem instance. This is achieved by training the policy in a budgeted multi-shot setting on a problem instance where past experiences are collected and stored in a memory. The memory is organised by nodes such that only information about a specific node is found in a data row that corresponds to that node. The policy retrieves data from the memory at each budget attempt and learns how to uses the data to decide on its next action.

The details of the MEMENTO training procedure are presented in Algorithm 1 and can be understood as follows. At each iteration, we sample a batch $B$ of instances from a problem distribution $\mathcal{D}$. Then, for each instance $\rho_i$ where $i \in 1, \ldots, B$, for $K$ budget attempts, we retrieve data from the memory. This is done as follows; given that the current selected node is $a_j$, we only retrieve data $M(a_j)$ associated with node $a_j$ and its starting point. However, the method is agnostic to starting point sampling and would work without it. The features associated with the retrieved action are then normalised. We add remaining budget as an additional feature and process the data by a Multilayer Perceptron (MLP) which outputs a scalar weight for each action. We compute correction logits by averaging the actions based on their respective weights. The correction logits are added to the logits of the base policy. We then rollout the resulting policy $\pi_{\tilde{\theta}}$ on the problem instance (i.e., generate a trajectory which represents a solution to the instance). After every policy rollout attempt, the memory is updated with transitions data such as the action taken, the obtained return, the log-probability of the action and the log-probability of the trajectory. These data are stored in the row that corresponds to the current selected node.

**Details about the loss** For each problem instance, we want to optimise for the best return. At each attempt, we apply the rectified linear unit (ReLU) function to the difference between the last return and the best return ever obtained. We use the rectified difference to compute the `REINFORCE` loss at each attempt to avoid having a reward that is too sparse and perform back-propagation through the network parameters of our model (including the encoder, the decoder and the memory networks). The sum of the rectified differences is equal to the best return ever obtained over the budget. As the budget is used, it becomes harder to improve over the previous best, the loss terms hence getting smaller. We found that adding a weight to the terms, with logarithmic increase, helped ensuring that the last terms would not vanish, and thus improved performance. We provide the mathematical formulation below.

Given a problem instance, we unroll $B$ trajectories that we store iteratively in the memory. Each trajectory $\tau_i$ generates a return $R(\tau_i)$. The advantage for each trajectory is defined as $\tilde{R}(\tau_i) = \max(R(\tau_i) - R_{best}, 0)$, where $R_{best}$ is the highest return found so far. The total loss for updating the policy is calculated using the REINFORCE algorithm: $\mathcal{L} = -\sum_{i=1}^{B} \log(1 + \epsilon + i)\tilde{R}(\tau_i) \sum_t \log \pi_M(a_t|s_t, M_t)$, where $\pi_M$ is the policy enriched with the memory $M_t$, using the logits $l_M$ defined in eq.1 in the paper. $\epsilon$ is a small number ensuring that the first term is not zero.

To keep the computations tractable, we still compute a loss at each step, estimate the gradient, and average them sequentially until the budget is reached, at which point we take a gradient update step and consider a new batch of instances.

In practice, we also observe that we can improve performance further by adding an optional refining phase where the base model is frozen, and only the memory module is trained, with a reduced learning rate (multiplied by $0.1$), for a few hours.

## C  HYPER-PARAMETERS

We report all the hyper-parameters used during train and inference time. For our method MEMENTO, there is no training hyper-parameters to report for instance sizes 125, 150, and 200 as the model used was trained on instances of size 100. The hyper-parameters used for MEMENTO are reported in Table 4. Since we also trained POMO and COMPASS on larger instances, we report hyper-parameters used for POMO and COMPASS in Table 5 and Table 6, respectively.

---

**Algorithm 1** MEMENTO Training

---

1: **Input:** problem distribution $\mathcal{D}$, problem size $N$, memory $M$, batch size $B$, budget $K$, number of training steps $T$, policy $\pi_\theta$ with pre-trained parameters $\theta$.
2: initialize memory network parameters $\phi$
3: combine pre-trained policy parameters and memory network parameters $\tilde{\theta} = (\theta, \phi)$
4: **for** step 1 to $T$ **do**
5:     $\rho_i \leftarrow \text{Sample}(\mathcal{D}) \ \forall i \in 1, \ldots, B$
6:     **for** attempt 1 to $K$ **do**
7:       **for** node 1 to $N$ **do**
8:         $m_j \leftarrow \text{Retrieve}\,(M) \ \forall j \in 1, \ldots, B$   {Retrieve data from the memory}
9:         $\tau_i^j \leftarrow \text{Rollout}\,(\rho_i, \pi_{\tilde{\theta}}(\cdot | m_j)) \ \forall i, j \in 1, \ldots, B$
10:       $m_j \leftarrow f(m_j, \tau_i^j)$   {Update the memory with transitions data}
11:       $R_i^* \leftarrow \max(R_i^*, \mathcal{R}(\tau_i^j)) \ \forall i \in 1, \ldots B$   {Update best solution found so far}
12:       $\nabla L(\tilde{\theta}) \leftarrow \frac{1}{B} \sum_{i \leq B} \texttt{REINFORCE}(\texttt{ReLU}(\tau_i^j - R_i^*))$   {Estimate gradient}
13:     $\nabla L(\tilde{\theta}) \leftarrow \frac{1}{K} \sum_{i=1}^K \nabla L(\tilde{\theta})$   {Accumulate gradients}
14:     $\tilde{\theta} \leftarrow \tilde{\theta} - \alpha \nabla L(\tilde{\theta})$   {Update parameters}

---

Table 4: The hyper-parameters used in MEMENTO

(a) TSP

| Phase | Hyper-parameters | TSP100 | TSP(125, 150) | TSP200 | TSP500 |
|---|---|---|---|---|---|
| Train time | budget | 200 | - | - | 200 |
| | instances batch size | 64 | - | - | 32 |
| | starting points | 100 | - | - | 30 |
| | gradient accumulation steps | 200 | - | - | 400 |
| | memory size | 40 | - | - | 80 |
| | number of layers | 2 | - | - | 2 |
| | hidden layers | 8 | - | - | 8 |
| | activation | GELU | - | - | GELU |
| | learning rate (memory) | 0.004 | - | - | 0.004 |
| | learning rate (encoder) | 0.0001 | - | - | 0.0001 |
| | learning rate (decoder) | 0.0001 | - | - | 0.0001 |
| Inference time | policy noise | 1 | 0.2 | 0.1 | 0.8 |
| | memory size | 40 | 40 | 40 | 40 |

(b) CVRP

| Phase | Hyper-parameters | CVRP100 | CVRP(125, 150) | CVRP200 | CVRP500 |
|---|---|---|---|---|---|
| Train time | budget | 200 | - | - | 200 |
| | instances batch size | 64 | - | - | 8 |
| | starting points | 100 | - | - | 100 |
| | gradient accumulation steps | 200 | - | - | 800 |
| | memory size | 40 | - | - | 40 |
| | number of layers | 2 | - | - | 2 |
| | hidden layers | 8 | - | - | 8 |
| | activation | GELU | - | - | GELU |
| | learning rate (memory) | 0.004 | - | - | 0.004 |
| | learning rate (encoder) | 0.0001 | - | - | 0.0001 |
| | learning rate (decoder) | 0.0001 | - | - | 0.0001 |
| Inference time | policy noise | 0.1 | 0.1 | 0.1 | 0.3 |
| | memory size | 40 | 40 | 40 | 40 |

Table 5: The hyper-parameters used in POMO

| Phase | Hyper-parameters | TSP500 | CVRP500 |
|---|---|---|---|
| Train time | starting points | 200 | 200 |
| | instances batch size | 64 | 32 |
| | gradient accumulation steps | 1 | 2 |
| Inference time | policy noise | 1 | 1 |
| | sampling batch size | 8 | 8 |

Table 6: The hyper-parameters used in COMPASS

| Phase | Hyper-parameters | TSP500 | CVRP500 |
|---|---|---|---|
| Train time | latent space dimension | 16 | 16 |
| | training sample size | 64 | 32 |
| | instances batch size | 8 | 8 |
| | gradient accumulation steps | 8 | 16 |
| Inference time | policy noise | 0.5 | 0.3 |
| | num. CMAES components | 1 | 1 |
| | CMAES init. sigma | 100 | 100 |
| | sampling batch size | 8 | 8 |

## D  MODEL CHECKPOINTS

Our experiments focus on two CO routing problems, TSP and CVRP, with methods being trained on two distinct instance sizes: 100 and 500. Whenever possible, we re-use existing checkpoints from the literature; in the remaining cases, we release all our newly trained checkpoints in the repository *anonymised for the review process*.

We evaluate MEMENTO on two CO problems, TSP and CVRP, and compare the performance to that of two main baselines: POMO (Kwon et al., 2020) and EAS (Hottung et al., 2022). The checkpoints used to evaluate POMO on TSP and CVRP are the same as the one used in Grinsztajn et al. (2023) and Chalumeau et al. (2023b), and the EAS baseline is executed using the same POMO checkpoint. These checkpoints were taken in the publicly available repository `https://github.com/instadeepai/poppy`. The POMO checkpoint is used in the initialisation step of MEMENTO (as described in Appendix B). To combine MEMENTO and COMPASS on TSP100, we re-use the COMPASS checkpoint available at `https://github.com/instadeepai/compass` and add the memory processing layers from the MEMENTO checkpoint trained on TSP100. This checkpoint is also released at *anonymised for the review process*.

For larger instances, we compare our MEMENTO method to three baselines: POMO, EAS and COMPASS. Since no checkpoint of POMO and COMPASS existed, we trained them with the tricks explained in Section 4. The process to generate the MEMENTO checkpoint and the MEMENTO (COMPASS) checkpoint is then exactly the same. All those checkpoints are available, for both TSP and CVRP.

## E  IMPLEMENTATION DETAILS

The code-base in written in JAX (Bradbury et al., 2018), and is mostly compatible with recent repositories of neural solvers written in JAX, i.e. Poppy and COMPASS. The problems' implementation are also written in JAX and fully jittable. Those come from the package Jumanji (Bonnet et al., 2023). CMA-ES implementation to mix MEMENTO and COMPASS is taken from the research package QDax (Chalumeau et al., 2023a). Neural networks, optimizers, and many utilities are implemented using the DeepMind JAX ecosystem (Babuschkin et al., 2020).

## F  CAN MEMENTO DISCOVER THE REINFORCE UPDATE?

In Section 3.2, we presented the architecture used by the auxiliary model that processes the memory data to derive the new action logits. The intuition behind this architecture choice is that it should be able to learn the REINFORCE update rule. Indeed the REINFORCE loss associated with a new transition is $R \log(\pi_\theta(a))$, such that $\frac{\partial R \log(\pi_\theta(a))}{\partial l_a} = R(1 - \pi_\theta(a))$. Therefore, simply having $H_{\theta_M}(f_a)$ match $R(1 - \pi_\theta(a))$ would recover a REINFORCE-like update. This is feasible, as $R$ and $\log(\pi_\theta(a))$ are included in the features $f_a$.

On Fig. 4, we compare the rule learned by MEMENTO compared to REINFORCE. In Appendix G, we present an ablation study of the features used in the update rule of MEMENTO, showing how much performance can be gained from the use of more information to derive the update rule.

## G  ABLATION STUDY: MEMENTO INPUT FEATURES

In Section 3.2, we present MEMENTO, in particular, we present all the inputs that are used by the neural module to derive the new action logits from the memory data. These inputs, or features, are information associated to each past action taken, and that help decide whether those actions should be taken again or not. In Section 4.1, we compare the rule learned by MEMENTO compared to the policy-gradient update REINFORCE. This comparison is made on the features that both REINFORCE and MEMENTO use: i.e. the action log probability, and the return (or advantage if a baseline is used). Although REINFORCE only uses those two features, MEMENTO uses more, which enables to refine even more the update, and also to adapt it to the budget remaining in the search process. As a recall, in addition to the log probability and return, MEMENTO uses: the log probability of the full trajectory, the log probability of the rest of the trajectory after the action was taken, the budget at the time the action was taken, the action logit suggested by MEMENTO when the action was taken, and the budget currently remaining.

To validate the impact of all the features used in the memory, we provide an ablation study of those features. To highlight the interest of using all the additional features, we report the performance of MEMENTO using only the return and the log probability, against the performance of MEMENTO with all features, on Fig. 8. We also report on Fig. 9 a bigger ablation where components are added one after the others.

We can extract two main observations: (i) first, adding the remaining budget completely changes the strategy. We can see on the right panel of Fig. 8 that having access to this additional feature enables MEMENTO to explore much more, and then to focus on high-performing solutions when it gets closer to the end of the budget; (ii) then, Fig. 9 confirms empirically that all features contribute to improving the overall performance.

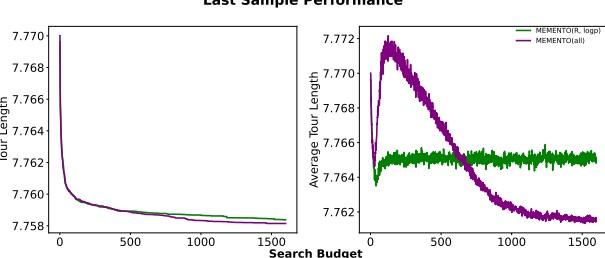

Figure 8: Ablation study of MEMENTO: comparing the impact of memory features that are not available in usual policy gradient estimations methods. The left plot reports the best solution found so far. The right plot shows the performance of the latest solution sampled. The plot illustrates how the additional features enable to achieve a complex exploration strategy, reaching a significantly more efficient adaptation mechanism.

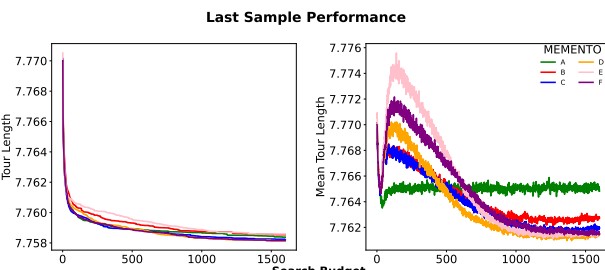

Figure 9: Full ablation study of MEMENTO: comparing the impact of memory features that are not available in usual policy gradient estimations methods. The left plot reports the best solution found so far. The right plot shows the performance of the latest solution sampled. A: return + logp; B: A + remaining budget; C: B + budget when action was taken; D: C + memory logp; E: D + trajectory logp; F: E + end trajectory logp.

## H    EVALUATION METRICS DURING TRAINING PHASE

In this section, we provide two plots of MEMENTO's training phase. They show the evolution of performance over time during MEMENTO's training on CVRP100. The left plot reports the evolution of the best tour length obtained during validation over time. The right plot reports the evolution of the improvement delta over time, i.e. the difference between the quality of the best solution generated minus the quality of the first solution generated. This metric shows well how the training phase results in MEMENTO learning an update rule that is able to significantly improve the base policy.

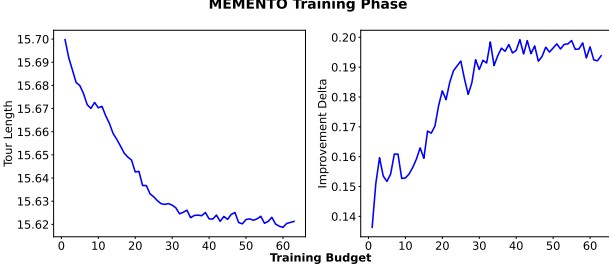

Figure 10: Evaluation metrics during the training phase of MEMENTO.

