# OpenReview forum: "Memory-Enhanced Neural Solvers for Efficient Adaptation in Combinatorial Optimization"
_ICLR.cc/2025/Conference — Submitted to ICLR 2025_

### Official Review · Reviewer_bp3q · 2024-10-21

**Soundness:** 2
**Presentation:** 3
**Contribution:** 2
**Rating:** 3
**Confidence:** 4

**Summary:**

The paper introduces MEMENTO, a novel memory-based approach designed to enhance existing constructive solvers for combinatorial optimization problems. MEMENTO leverages information from past solution attempts to improve the construction process, using a multilayer perceptron (MLP) network that takes features such as past actions, log-likelihood, and remaining search budget as input. This information is used to adjust the solution generation policy during inference, leading to higher-quality solutions within the given computational budget. The authors demonstrate that MEMENTO can improve the performance of base constructive models like POMO or COMPASS on problems such as the Traveling Salesman Problem (TSP) and Capacitated Vehicle Routing Problem (CVRP).

**Strengths:**

Adapting the solution generation process using memory to account for previous attempts is undoubtedly a valuable and significant research direction.

**Weaknesses:**

This paper introduces a 'meta-learning' approach in building solvers for combinatorial optimization problems, structured around two levels of machine learning. The lower-level learning takes place during inference, where previous solution attempts stored in memory are used to update the node selection policy. The upper-level (meta) learning, referred to as 'training' in the paper, involves training a multilayer perceptron (MLP) to find the optimal parameters that shape the behavior of the lower-level learning.

For the lower-level learning, I find the proposed method overly simplistic, with two primary issues.

First, the method indiscriminately utilizes all data in memory associated with the current node. Past experiences at the same 'current node' should not automatically be considered as occurring in the same 'state.' Although I haven’t thoroughly verified this, I believe the current implementation doesn’t necessitate storing the entire raw history. Instead, it could maintain only the accumulated 'logit correction values' and update them as new data arrives. In this sense, the term 'memory' may be too generous for the proposed approach, which feels closer to a simple bookkeeping method (EAS-tab) or basic active learning, both of which adjust the policy after each trajectory rollout iteration. A more effective memory system would incorporate mechanisms to retrieve the most relevant and important data while discarding irrelevant or potentially harmful data.

Second, the input features chosen for learning appear somewhat arbitrary. While the inclusion of the 'budget' feature is a helpful addition, many of the other features seem to offer limited value (as shown in Figure 9), and there is no clear theoretical basis for their usefulness. Moreover, generalizing these features to other combinatorial problems beyond fixed-sized routing may prove challenging.

Regarding the upper-level learning, the authors should provide a more precise and detailed explanation of their meta-learning approach, as applying meta-learning to combinatorial problems is an especially difficult task. To find a globally optimal solution using a reinforcement learning (RL) approach, an end-to-end method may be more suitable than optimizing over a few grouped trajectories, as done in this paper. To partly address this limitation, the authors manually adjust the learning process by logarithmically increasing the loss weight. An effectively designed RL approach would ideally discover optimal weights autonomously, eliminating the need for manual adjustments.

In addition to these concerns, the proposed method yields only marginal improvements over other baselines. Consequently, I believe the paper does not meet the high standards of ICLR.

**Questions:**

I would like to understand the MLP structure used by the authors, described in Table 4.

Does "memory size 40, number of layers 2, hidden layers 8" imply the following MLP structure?

1) An input layer with 40 neurons
2) A first hidden layer with 8 neurons
3) A second hidden layer with 8 neurons
4) An output layer with 1 neuron

---

> ### Author Response · Authors · 2024-11-23
>
> > W1: Node retrieval strategy.
>
> We fully agree that the same node does not induce the same state and that ideally we would retrieve data based on similarity to the current partial solution. But this comes with a computation cost since it requires to get a similarity measure between the current solution and all the data points in the memory and then to extract the k most similar points. We observed that retrieving from the same node was an excellent proxy for similarity, and that the most similar points are very likely to come from the same node. This retrieval strategy hence provided a better tradeoff between quality and computation cost. This choice is discussed in the paragraph “retrieving data from the memory” (l. 200) but we are ready to extend it or to add details in the appendix if deemed necessary.
>
> Please refer to Q1 from Reviewer YhmS for details about an alternative retrieval mechanism we actually ended up discarding in favor of our node retrieval.
>
> > W2: The current implementation could maintain only the accumulated 'logit correction values' and update them as new data arrives. The proposed approach feels closer to a simple bookkeeping method (EAS-tab).
>
> Our method already has a similar behavior since the logit correction is stored with new data points. Hence at the next iteration the MLP can simply re-use the information as desired, or discard it (by using the “age” of the data, which tells how much of a shift might have happened). Having the capacity to discard it is a desired property since there is a policy shift happening.
>
> The simple book-keeping method EAS-Tab uses a heuristic (“requiring deeper understanding of the CO problem”) to reinforce the probability to replicate the best found solution whereas our method uses a fully learned rule and can use 40 past solutions to inform the next iteration.
>
> > W3: Motivation for the memory input features, and generalization to other problems.
>
> In section 3.2 and appendix F,  we elaborate on the choice of those features. “Log probability” and “Return” are the core elements of a REINFORCE policy update. The “Log probability suggested by the memory” enables the accumulation of logit corrections over time. The “budget at the time the solution was built”, which can also be seen as the age of the data, enables to account for the distribution shift happening in the memory, i.e. giving more importance to recent data. The “Log probability of the entire trajectory” can be used to avoid discouraging an action that brought low return if that action was part of a trajectory that is believed to be unpromising anyway.
>
> Those features are agnostic to the problem at hand, it hence should not be challenging to use them beyond TSP and CVRP. The only problem-size related one being the “Log probability of the entire trajectory”, which can either be normalized, or simply removed from the features. We can add those elements in the appendix of the paper.
>
> > W4: To find a globally optimal solution using a reinforcement learning (RL) approach, an end-to-end method may be more suitable than optimizing over a few grouped trajectories.
>
> The end-to-end approach would hardly work in this NCO setting. This would require learning how to extract relevant information from past attempts, in addition to learning how to re-use this information. The compute cost of the “extraction step” would typically be an order of magnitude more than the typical decoding step. At training time, it is unclear whether the computation graph would fit in memory, or would allow a big enough batch size to get stable gradients. At inference time, the method would be one or two orders of magnitude slower than the concurrent methods on the standard benchmark, and likely to have poor scaling properties.
>
> Our final approach results from a compromise between learning as much as we can, and staying within the computation constraints imposed by the typical tasks that are considered in the NCO field currently.
>
> > W5: The proposed method yields only marginal improvements over other baselines.
>
> For the sake of space, we refer the reviewer to our answer to W1 of Reviewer Sd71.
>
> > Q1: I would like to understand the MLP structure used by the authors, described in Table 4.
>
> We apologize for the misprint in our hyperparameters table, “hidden layers” must be replaced by “hidden units”. To understand the MLP structure, let’s detail the process. Each memory entry has 7 associated features, which are processed by the MLP. The two hidden layers have 8 units. The output has a size of 1.
>
> Hence the size of the MLP is [7, 8, 8, 1]. If the memory size chosen is 40, then the 40 entries are processed in parallel.. This is illustrated on Fig. 2 and described in the paragraph “processing data to update actions” of Section 3.2.

---

> > ### Comment · Reviewer_bp3q · 2024-11-26
> >
> > > W1: Node retrieval strategy
> >
> > I understand that using the same node retrieval strategy was the best choice among the practical options. However, this approach limits the applicability of the proposed method to other general combinatorial optimization problems.
> >
> >
> > > W2: accumulation vs memory
> >
> > I now understand that the proposed method only utilizes recent history and discards older data. This distinguishes this method from others I previously mentioned. It would be nice if there is an ablation test on the effect of this "discarding" strategy.
> >
> >
> >
> > > W3: input features
> >
> > I still believe that Figure 9 illustrates how most of the input features do not significantly contribute to the overall quality of the solver's performance.
> >
> >
> > > W4: meta-learning
> >
> > The paper does not place much emphasis on it being a meta-learning study. A hand-crafted meta-learning stretagy is used, and training a [7, 8, 8, 1] MLP appears to be a less compelling task.
> >
> > I will keep my original rating.

---

> > > ### Author Response · Authors · 2024-11-27
> > >
> > > We thank the reviewer for their additional comments.
> > >
> > > > This approach limits the applicability of the proposed method to other general combinatorial optimization problems.
> > >
> > > This retrieval strategy can be used for other problems. The practitioner can also tune the retrieval strategy if they want: this is an implementation detail that does not limit the applicability of the MEMENTO framework.
> > >
> > > > Impact of input features.
> > >
> > > The tour length performance gives the impression that the input features have very little influence. Nevertheless, one should keep in mind that (i) we are very close to optimality, hence performance improvements are tough to obtain, and small values are not insignificant (ii) the right panel – which reports the tour length obtained per batch of attempts – shows that each input feature does impact the search strategy.
> > >
> > > Concretely, we suggest two modifications: (i) we can report the tour length performance on a log scale to help visualise the performance difference close to optimality on the right panel of Fig. 9 (ii) we can make it clear in the paper that the exact choice of features is an implementation choice within the framework, and that practitioners can stick to a subset of the analysed features at their own discretion.
> > >
> > > > The paper does not place much emphasis on it being a meta-learning study
> > >
> > > This paper is not a meta-learning study. This paper introduces a framework to improve the adaptation of neural solvers at inference time. It identifies limitations of policy-gradient updates within the context of inference time with multiple shots, and proposes to learn a policy update to overcome those limitations. Since it learns a policy update, there is a connection with meta-learning, and consequently, we propose an analysis similar to other work in meta-learning (Fig. 4). We can add an appendix to discuss some other meta-learning approaches that we discarded.
> > >
> > > > A hand-crafted meta-learning strategy is used, and training a [7, 8, 8, 1] MLP appears to be a less compelling task.
> > >
> > > The task is to learn a policy update rule, the MLP should be designed for this task, not to “appear compelling”. We do not see any reason to use a large and complex architecture when a small one has enough capacity to learn the right update rule. Plus, a large MLP comes with a computation cost.

---

> > > > ### Comment · Reviewer_bp3q · 2024-11-28
> > > >
> > > > > Impact of input features
> > > >
> > > > I would like to suggest that Figure 9 can be drawn for problems that are not so close to be solved to optimality, e.g. CVRP.
> > > >
> > > >
> > > > > Meta-learning
> > > >
> > > > When considering this paper not as a meta-learning study but as an inference technique proposal, the definition of the proposed framework becomes somewhat ambiguous.
> > > >
> > > > If we focus on improving solutions solely at "inference time" with a pre-trained model (like POMO-trained model), MEMENTO requires additional training time before it can be applied to a new set of problems. In contrast, all other baseline methods compared in Figure 3 can be applied almost directly. This makes the comparison unfair.
> > > >
> > > > Alternatively, if we consider a scenario where there's sufficient time for preparation before inference (allowing MEMENTO to train), then demonstrating "Generalization" performance has little meaning. In this case, practitioners now have the option to train a model specifically for the new problem set, possibly by fine-tuning the original model. Therefore, in this scenario, MEMENTO should be compared against "fine-tuned models" for a fair assessment.

---

> > > > > ### Author Response · Authors · 2024-11-28
> > > > >
> > > > > > suggestion for Fig. 9
> > > > >
> > > > > We thank the reviewer for the suggestion. We agree that the ablation study will carry even more insights that way. Additionally, having the ablation study on both environments will strengthen findings.
> > > > >
> > > > > > inference technique proposal.
> > > > >
> > > > > The usual setting in NCO is a two-phase process, with different assumptions. First, the training phase, where time and computation are largely available, but the inference distribution is unknown. Then, the inference phase, where the new instances are revealed; but the budget (time and hardware, hence attempts) is usually much more constrained.
> > > > >
> > > > > > training phase fairness
> > > > >
> > > > > Hence, for the training phase to be fair, we need to (i) make sure all methods can be trained as much as they need to, i.e. till convergence – which is the case for all methods compared here – (ii) make sure that the training distribution used does not create unfair bias – which is the case here since all methods are trained on the same distribution, i.e. TSP (resp. CVRP) 100 –.
> > > > >
> > > > > Although it is true that MEMENTO requires additional training, POMO and other methods cannot make use of any additional training time, although it is also available to them. For this reason, the training phase of this study is fair.
> > > > >
> > > > > > inference phase fairness
> > > > >
> > > > > Even though there is "sufficient time for preparation before inference", the new problem set is never revealed before the beginning of the inference phase. We want to clarify that MEMENTO is always only trained on the original training distribution, just like other methods; and none has access to the inference distribution before the beginning of the inference phase. Hence, no model can be fine-tuned on the inference-time distribution. Consequently, we are confident that our assessment of the inference time performance is fair.
> > > > >
> > > > > All in all, our NCO evaluation pipeline is coherent with the literature, and aligned with settings that would be observed in real world scenarios. We are hence confident that we provide a fair ground for comparing the methods.

---

### Official Review · Reviewer_R4rn · 2024-11-03

**Soundness:** 2
**Presentation:** 4
**Contribution:** 3
**Rating:** 5
**Confidence:** 4

**Summary:**

This paper proposes MEMENTO, a method for fast instance-specific adaptation in CO problems when a certain budget is allowed by utilizing a memory module. The proposed memory module is a lookup table storing information encountered during the online search process that is important for decision-making, such as the outcome of certain actions. This is then fed into an MLP, whose output modifies the action probabilities at each step. The method is evaluated in standard routing problems TSP and CVRP up to 500 nodes where it demonstrates SOTA results against RL-trained autoregressive solvers for CO.

**Strengths:**

1. The paper is very well written and clear with relevant citations to previous literature.

2. The proposed MEMENTO module is novel and simple enough to be applied to a range of autoregressive CO solvers in the future, and thus, I believe it is an available addition to the NCO community.

3. Good overall performance (albeit with some concerns about baselines below) and experimental validation, including classical benchmarking, zero-shot combination with new solvers, and scaling to large sizes.

4. Code is provided, and authors make an effort to make their checkpoints available to the community.

**Weaknesses:**

1. My biggest concern is about fairness in comparison with EAS. In particular, the values reported in the paper are worse than the ones in the original paper. For instance, for the Kool et al. (2019) 10k instances with 100 nodes, the value reported is 7.778 vs the original 7.769 for the TSP (MEMENTO: 7.768) and 15.66 vs 15.63 for the CVRP (MEMENTO: 15.65). Compared to the values reported in the original paper, MEMENTO would be worse than EAS. This holds true at larger sizes too. Do the authors have an explanation for this?

2. MEMENTO is only applied to routing problems despite the title appealing to a broader “Combinatorial Optimization”. In this sense, experimenting with differently structured environments such as the Job Shop Scheduling (JSSP) as done in COMPASS and EAS would be beneficial.

3. When computing gaps, it would be best to do so compared to SOTA heuristic methods. For instance: in Figure 7, in Table (b), only LKH3 is reported, while HGS is much more powerful on CVRP. However, the authors do report HGS in the appendix, which obtains much better solutions than LKH. This also applies in particular to Table 1, where the “optimality gap” appears to be negative.

4. Given that only routing problems are considered in this paper, it would be beneficial to mention the additional work [1].

5. Some questions regarding hyperparameters remain, see below questions.


---

[1] Son, Jiwoo, et al. "Meta-sage: Scale meta-learning scheduled adaptation with guided exploration for mitigating scale shift on combinatorial optimization." International Conference on Machine Learning. PMLR, 2023.

**Questions:**

1. What is the impact of the MLP in terms of cost? Since this has to be called each time, I wonder whether similar results could be obtained with a simple linear layer as well.

2. In Figure 2: MEMENTO’s logit update encourages with higher amplitude high-reward actions rather than low-return ones. Is this due to the `ReLU` applied to the reward, constraining it to be strictly positive? I wonder if this analysis would hold without such constraint.

3. Can your method be applied to broader problems that include e.g. edge features as the JSSP?

4. What is the impact of the memory size? Would increasing it be beneficial?

5. MEMENTO appears to be >$2\times$ slower than EAS at larger sizes as seen in Appendix A.1. Would EAS, provided with as much time budget as MEMENTO, eventually surpass the latter?

---

> ### Author Response · Authors · 2024-11-23
>
> > W1: Confusion with results reported in [1].
>
> The discrepancy between those results comes from the use of problem augmentation using symmetries. It is a problem-specific trick that can only be used for a few CO problems. Additionally, most prior work assumes that this additional x8 batching can be achieved seamlessly, which is very unlikely in practice, when simulating complex real world scenarios.
>
> Using this trick means decreasing the room for search and adaptation, since 87.5% of the budget is consumed to squeeze performance through uncontrolled network variance, rather than letting methods use principled strategies. Consequently, we genuinely believe that the NCO community would benefit from using a default evaluation setting that does not rely on this trick.
>
> > W2: MEMENTO is only applied to routing problems despite the title appealing to a broader “Combinatorial Optimization”.
>
> We agree that adding another benchmark would strengthen the paper. Nevertheless, we decided to rather scale those environments to 500 nodes, which had never been done before and brought very interesting results. This effort was costly since we scaled all POMO, EAS, COMPASS and MEMENTO; and should be valuable to the community since we found unexpected rankings, plus we release the weights. We believe that this effort brings enough empirical evidence and justifies leaving other environments for future work. Since the design of our method makes it applicable to other COPs, we would like to keep the title that way, but are ready to reconsider this if this opinion is shared amongst all reviewers.
>
> > W3: Compute optimality gaps in comparison to SOTA heuristic methods, particularly HGS on CVRP large-scale.
>
> We kept LKH3 as a reference since most of the methods we compare to report results with LKH3, and optimality gap cannot always be recomputed since only truncated results are given. We stayed consistent when moving to large-scale experiments. We are ready to update this if deemed necessary amongst the reviewers.
>
> > W4: Given that only routing problems are considered in this paper, mention additional work [3].
>
> We thank the reviewer for pointing out recent work Meta-SAGE, which we will add to the related work section.
>
> > Q1: What is the impact of the MLP in terms of cost? Could it be replaced by a linear layer?
>
> The MLP we use has 2 layers, with 8 hidden units. The computation has to be made for each memory entry sampled, which is 40 in most of our experiments. Those computations are done in parallel. Interestingly, this does not depend on the base architecture of the policy, hence this cost becomes negligible compared to the forward pass of the base policy when its size increases.
>
> We have obtained better performance with a MLP compared to a linear layer, which we believe is explained by the fact that most rules we want to learn (i) require non-linearities (ii) can require interactive terms. Fig. 4 confirms that the learned rule is non-linear.
>
> > Q2: MEMENTO’s logit update encourages, with higher amplitude, high-reward actions rather than low-return ones. Is this due to the ReLU applied to the reward, constraining it to be strictly positive? Would this analysis hold without such constraint?
>
> It is indeed linked to the ReLU used in the reward and it is a desired property. Removing the ReLU would mean optimizing for the average score obtained amongst the attempts rather than optimizing for the maximum.
>
> > Q3: Can your method be applied to broader problems?
>
> Yes, MEMENTO’s pipeline is fully agnostic to the COP. The practitioner can tune the data retrieval strategy or add problem-specific information in the data features, but this is not a requirement for MEMENTO to work.
>
> Interestingly, most concurrent methods that can be applied to TSP 500 (DIMES, MOCO, etc…) cannot be applied to COPs that cannot be expressed as heatmap; making them less applicable than MEMENTO.
>
> > Q4: Would EAS, provided with as much time budget as MEMENTO, eventually surpass the latter?
>
> This really depends on the given problem, and time budget given, but also on the hardware accessible. For some of the experiments reported in the paper, it would likely be the case (for instance CVRP 125), but many others would not be affected (e.g. TSP results).
>
> Additionally, the scaling laws reported on Fig. 7 show that for larger instances and larger networks, MEMENTO actually becomes faster than EAS, by a significant margin.

---

> > ### Comment · Reviewer_R4rn · 2024-11-25
> >
> > I thank the authors for their answers, here are some follow-up remarks:
> >
> > > The discrepancy between those results comes from the use of problem augmentation using symmetries. It is a problem-specific trick that can only be used for a few CO problems. Additionally, most prior work assumes that this additional x8 batching can be achieved seamlessly, which is very unlikely in practice, when simulating complex real world scenarios.
> >
> > I agree with your assessment that using this augmentation trick should not be the main point of a method. However, since EAS was proposed using it, I believe just showing that MEMENTO+trick > EAS+trick would suffice because, at the moment, it seems that EAS+trick > MEMENTO. Also, although the trick is problem-specific, the studied problems are Euclidean routing only, where the trick can be applied. Would MEMENTO with the trick work better than EAS with it?

---

> > > ### Author Response · Authors · 2024-11-26
> > >
> > > We thank the reviewer for acknowledging our answers and engaging further in the discussion.
> > >
> > > > At the moment, it seems that EAS+trick > MEMENTO
> > >
> > > Although EAS + trick is indeed outperforming MEMENTO, we can also observe that POMO + trick is outperforming EAS on TSP (from the results reported by COMPASS and EAS). Obviously, we should not deduce that sampling is better than EAS. Hence, we cannot conclude anything from the fact that EAS + trick outperforms MEMENTO.
> > >
> > > > Since EAS was proposed using it, I believe just showing that MEMENTO + trick > EAS + trick would suffice because, at the moment, it seems that EAS + trick > MEMENTO.
> > >
> > > The augmentation was actually not introduced along EAS fine-tuning method. It was introduced in POMO, and EAS is fully agnostic to this trick, and probably just inherited it when evaluating their method. We are not saying that the augmentation trick is the main point of EAS (nor POMO), we rather support that it is not the best way to compare current and future methods. A good setting to compare inference-time strategies should have two main properties: first, help emphasize the adaptation and search capacity of the compared methods;  second, correspond to a budget setting that is feasible in practice.
> > >
> > > Concerning the first point, our concern is that the trick consumes 87.5% of the budget, hence only little credit can be given to the actual efficacy of the adaptation/search methods we are trying to evaluate. We believe that a stronger conclusion can be drawn from comparing without trick than with trick.
> > >
> > > Concerning the second point, we believe that the standard benchmark with the augmentation trick leads to a setting that is very unlikely in practice. The benchmark uses the “starting point trick” (also introduced by POMO), i.e. each rollout is actually the best solution from rollouts obtained from each possible starting point. Hence, in TSP200, each attempt is already corresponding to 200 attempts. When using the “augmentation trick”, this means we are now rolling out 1600 episodes in parallel. Although this is still possible with small instance size and simple environments, this becomes absolutely intractable for larger instance size, but also for environments that are complex to simulate.
> > >
> > > We have actually tried to use large batch sizes for our experiments on TSP and CVRP 500 that are reported in section 4.3, and it was already challenging to use a method like EAS with 120 parallel attempts because the back-propagation becomes too costly; hence 1600 parallel attempts is not an option.
> > >
> > > A current issue in the NCO literature is that methods are often compared with a single budget setting (a single and fixed number of attempts, described in a number of sequential attempts, and a number of attempts achieved in parallel at each sequential step). We try to address this by comparing EAS and MEMENTO on a range of settings (Fig. 6), which enables us to reach more nuanced conclusions: the ordering of methods might often depend on the budget, and what is really important for a practitioner is to know when to use each method.
> > >
> > > We do not discard the use of a standard benchmark with fixed budget since it enables to give a reference to assess methods on, and it makes it easier to report results of a set of methods; but we believe that this benchmark should at least use a setting that is feasible in practice.
> > >
> > > For all those reasons, we believe that the comparative results between MEMENTO and EAS without the trick carry more meaning than would a comparison that uses the trick.

---

> > > > ### Comment · Reviewer_R4rn · 2024-11-26
> > > >
> > > > Thanks for your answer.
> > > >
> > > > Again, while I agree that not using the trick for both methods can be a fair setting. However, I still believe EAS and MEMENTO should be compared with the trick.
> > > >
> > > > There are two reasons why I believe a comparison should be made:
> > > > 1. To beat the original results of EAS
> > > > 2. EAS hyperparameters were tuned under the augmentation trick assumption. If the augmentation trick is not used, its hyperparameters should be tuned again
> > > >
> > > > Finally, I checked the supplementary material under `memento/environments/cvrp/utils.py` where the `get_augmentations` function (i.e. the x8 augmentation trick) was already implemented. As such, I believe there should be no issue showing MEMENTO+trick > EAS+trick.

---

> > > > > ### Author Response · Authors · 2024-11-28
> > > > >
> > > > > We thank the reviewer for their additional comments.
> > > > >
> > > > > > Comparing with the trick to beat EAS on the original results
> > > > >
> > > > > Our objective is to beat EAS on a setting that is appropriate and feasible in practice, more than on the original setting. Plus, we already provide a study of both methods on a larger set of budget (Fig. 6). We will nevertheless try our best to provide these additional experiments before the end of the discussion, and trust that the forthcoming results will be interpreted within this broader context – in particular keeping in mind the relative relevance of the budget settings –.
> > > > >
> > > > > > EAS hyper-parameters
> > > > >
> > > > > Concerning the hyper-parameters, EAS paper does not provide hints on how to adapt the hyper-parameters – plus, the values used are not in the paper and can only be found by searching the codebase –. We nevertheless saw that they keep the same hyperparameters for all TSP sets they evaluate on, and consequently do not change those with respect to the parallel batch size (TSP 100 rolls out 800 solutions in parallel, TSP 200 rolls out 1600, but the same hyper-parameters are used). We hence believe to be giving a fair effort with respect to the information that is available to us.

---

> > > > > > ### Author Response · Authors · 2024-11-29
> > > > > >
> > > > > > We have obtained the results with augmentations on the TSP benchmark. We observe that MEMENTO outperforms EAS on three of the four tasks:
> > > > > >
> > > > > > | Method | TSP 100 | TSP 125 | TSP 150 | TSP 200 |
> > > > > > |----------|----------|----------|----------|----------|
> > > > > > | EAS    | 7.768     | 8.590     | 9.361     | **10.730**     |
> > > > > > | **MEMENTO**    | **7.765**     | **8.586**     | **9.355**     | 10.743     |
> > > > > >
> > > > > > We are currently running the CVRP benchmark with augmentations and will share the results with the reviewer as soon as the experiments are finished.

---

> > > > > > > ### Author Response · Authors · 2024-12-02
> > > > > > >
> > > > > > > Below are the results for the CVRP benchmark with augmentations:
> > > > > > >
> > > > > > > | Method | CVRP 100 | CVRP 125 | CVRP 150 | CVRP 200 |
> > > > > > > |----------|----------|----------|----------|----------|
> > > > > > > | EAS    | 15.623     | **17.473**     | **19.261**     | 22.556     |
> > > > > > > | MEMENTO    | **15.616**     | 17.511     | 19.316     | **22.515**     |
> > > > > > >
> > > > > > > We can observe that MEMENTO outperforms EAS in distribution (CVRP 100), and for the large instances task (CVRP 200). EAS leads on the two other tasks. Note that we have not tuned MEMENTO's hyper-parameters for these runs.
> > > > > > >
> > > > > > > MEMENTO leads the augmented benchmark overall: it consistently leads in-distribution, and both methods are competing closely out-of-distribution. Those results do not fundamentally change the main message of our paper since our in-depth analysis (Fig. 6) with different budget settings already shows nuanced conclusions for high parallelism, which is the case here since using augmentations induces rolling out between 800 and 1600 solutions in parallel at each batch of attempt.
> > > > > > >
> > > > > > > We hope that this additional benchmark addresses the reviewer's concerns regarding our results and confirms the strength of the comparison provided.

---

> > > > > > > > ### Comment · Reviewer_R4rn · 2024-12-02
> > > > > > > >
> > > > > > > > I thank the authors for their additional results.
> > > > > > > >
> > > > > > > > From my understanding, MEMENTO performs better than EAS in distribution. Out-of-distribution, it is pretty much random (in TSP: better on a smaller scale, worse at large; the opposite is true in CVRP).
> > > > > > > >
> > > > > > > > Nonetheless, I do appreciate the authors for reporting the results in a fairer setting.
> > > > > > > >
> > > > > > > > I have one last question: Why do the results for CVRP 200 differ from those in the EAS paper? Also, in Figure 3 (b), the reported results appear to be better than the results with augmentation that you just reported, while the opposite would be expected.

---

> > > > > > > > > ### Author Response · Authors · 2024-12-02
> > > > > > > > >
> > > > > > > > > We thank the reviewers for acknowledging the new results.
> > > > > > > > >
> > > > > > > > > > MEMENTO performs better than EAS in distribution. Out-of-distribution, it is pretty much random.
> > > > > > > > >
> > > > > > > > > We agree with these observations on the benchmark with augmentation (where MEMENTO was not tuned, and that we still believe to not be the most appropriate way to compare methods). Concerning the overall conclusion, if we aggregate the results on the standard benchmark with and without augmentations, MEMENTO leads 4 out of 4 in-distribution tasks; and leads 9 out of 12 out-of-distribution tasks. The empirical results out-of-distribution hence still lean towards MEMENTO.
> > > > > > > > >
> > > > > > > > > > Why do the results for CVRP 200 differ from those in the EAS paper?
> > > > > > > > >
> > > > > > > > > The difference in the results come from the JAX implementation of EAS. We re-use the implementation published with COMPASS. They disclaim this discrepancy in Appendix N. From this appendix, it seems that (i) the JAX implementation is significantly faster than the original PyTorch implementation (ii) it outperforms or equals the original performance on 6 of the 8 tasks (iii) but cannot match it on 2 tasks. This JAX implementation enabled us to conduct numerous additional comparisons: comparison without augmentation, comparison on the large set of possible budget, scaling EAS to TSP and CVRP 500 (which had never been done before), and comparison based on the runtime (see answers to reviewer YhmS). We keep it as the unique implementation for our entire study for consistency and coherence.
> > > > > > > > >
> > > > > > > > > > in Figure 3 (b), the reported results appear to be better than the results with augmentation
> > > > > > > > >
> > > > > > > > > When using augmentations, the "sequential budget" is used in "parallel budget". In CVRP 200 with augmentations, the methods have 200 sequential batch of attempts, where each batch is 1600 attempts in parallel (each 200 starting point, each 8 augmentation). Without augmentations, the methods have 1600 sequential batch of attempts, where each batch is 200 attempts (each 200 starting point, no augmentation). This is akin to what is done in COMPASS.
> > > > > > > > >
> > > > > > > > > This means the budget is either used for the "network variance"-based search, or for more sequential attempts that enable the methods to adapt (with gradient descent for EAS, with memory and action distribution update for MEMENTO). Since the base policy is trained on CVRP 100, doing high-quality inferences on CVRP 200 requires a lot of policy adaptation: the "network variance" search provided by the augmentation trick lacks efficacy, and hence gets outperformed by the principled search of both methods. This explains why both MEMENTO and EAS get better results on CVRP 200 without augmentation.

---

> > > > > > > > > > ### Comment · Reviewer_R4rn · 2024-12-03
> > > > > > > > > >
> > > > > > > > > > >  This explains why both MEMENTO and EAS get better results on CVRP 200 without augmentation.
> > > > > > > > > >
> > > > > > > > > > Still, EAS-emb in the original paper reports a 0.88% gap regarding the same LKH3 result, while you report 2.46% (without augmentation), and with augmentation, it is even worse somehow. It does not seem to me that the results were successfully reproduced.
> > > > > > > > > >
> > > > > > > > > > > the "network variance" search provided by the augmentation trick lacks efficacy, and hence gets outperformed by the principled search of both methods.
> > > > > > > > > >
> > > > > > > > > > It might, but in distribution (and also close to it), the trick in fact produces better results. Given their variability, I am unsure how many conclusions can be drawn from this.
> > > > > > > > > >
> > > > > > > > > > Overall, I am still unconvinced regarding these results, and I am inclined to keep the current score. While I thank the authors for clarifying some details, reproducibility and fair comparisons were only partially addressed. I invite the authors to improve the paper in these aspects.

---

> > > > > > > > > > > ### Author Response · Authors · 2024-12-04
> > > > > > > > > > >
> > > > > > > > > > > > results reproducibility
> > > > > > > > > > >
> > > > > > > > > > > This JAX EAS implementation improves upon the original one on 5 out of 8 tasks. It is open-sourced, and its results have been reported in a Neurips paper (hence peer-reviewed). The code and the checkpoint – that we have simply reused – are fully accessible online, hence enabling transparency.
> > > > > > > > > > >
> > > > > > > > > > > > I am unsure how many conclusions can be drawn from this
> > > > > > > > > > >
> > > > > > > > > > > We agree that the standard benchmark has limitations and cannot be used alone to provide a thorough comparison of methods. To address this, we have expanded the empirical comparisons by evaluating all methods on larger instances (TSP and CVRP 500), and are releasing the weights to enable other researchers to build upon this. We regret that this contribution has not received more attention in the overall rebuttal discussions.
> > > > > > > > > > >
> > > > > > > > > > > While we may not fully agree with some of the reviewer's conclusions, we deeply value their thoughtful engagement and dedication throughout the discussion period. Their insights will help guide improvements in the updated manuscript.

---

### Official Review · Reviewer_Sd71 · 2024-11-04

**Soundness:** 2
**Presentation:** 1
**Contribution:** 2
**Rating:** 3
**Confidence:** 3

**Summary:**

This paper introduces a fine-tuning method applied during inference to enhance construction methods for combinatorial optimization. It stores historical trajectories from fine-tuning as memory, which is processed by an MLP to adjust the original action probabilities. New solutions are then sampled from this memory-augmented distribution. The pretrained model and memory network are updated using an improvement reward based on the difference between the current solutions and the best-so-far solutions. Experiments are conducted on TSP and CVRP with problem sizes of 100 (generalizing to 125, 150, and 200) and 500.

**Strengths:**

1. The source code is provided.
2. This work has the potential to enhance the pre-trained construction methods for TSP and CVRP.
3. The idea of reusing the historical trajectories (i.e. the memory) is interesting.

**Weaknesses:**

1. Marginal Improvement: The improvement of MEMENTO over EAS appears marginal, especially when generalizing to larger scales (e.g., TSP200 in Figure 3). In the CVRP results, the improvement of MEMENTO is also not significant.
2. Scalability Concerns: In the larger-scale experiment (n=500), MEMENTO only slightly outperforms COMPASS, while introducing higher computational overhead.
3. Incomplete Literature Review: The literature review lacks coverage of works focused on scalability and generalization.
4. High Computational Cost of Fine-Tuning: The proposed fine-tuning method introduces additional computational overhead.
5. Missing Generalization Experiments: Would be useful to add some generalization experiments on different distributions.
6. Writing Quality: The writing lacks logical flow, and the table formatting needs improvement.

**Questions:**

1. Why does POMO with sampling strategies perform worse than POMO with the greedy rollout when generalizing to N=200 in Figure 3?
2. Would it be feasible to apply MEMENTO to improvement methods as well? If so, are there any considerations that would impact its performance?
3. What's the inference time of the methods displayed in Figure 3? The bar chart results seem to replicate those in the table on the left, making it feel a bit redundant
4. Can MEMENTO be applied to other COPs? For example, I noticed that the code includes a preliminary implementation for the Knapsack Problem. I’d be interested to see the results and understand how MEMENTO might extend to different COPs.

---

> ### Author Response · Authors · 2024-11-23
>
> > W1:  The improvement of MEMENTO over EAS appears marginal.
>
> The improvement of MEMENTO over EAS can appear marginal given how close methods are to optimality, in particular on the standard TSP/CVRP-[100-200] benchmark, but is still significant and in the typical magnitude observed in published NCO methods, and consistent over 11 of the 12 tasks. Furthermore, the standard benchmark uses a budget introduced by EAS, but Fig. 6 shows that for a wide range of other budgets, the gap between those methods can actually increase by a large margin.
>
> Finally, a key message from our paper is that there is not any single NCO approach that is uniformly best; rather it depends on the setting (e.g. one vs few vs many shots, sequential solution attempts vs parallel decoding etc).
>
> > W2: Slight improvements of MEMENTO over COMPASS on larger-scale with a higher computational overhead.
>
> In all the large-scale experiments, adding MEMENTO over COMPASS provides more than 10% decrease of the gap to the reference industrial solver, which corresponds to a significant improvement in the field of NCO.
>
> > W3: The literature review lacks coverage of works focused on scalability and generalization.
>
> We have focused our literature review on the most important context of our contribution, while trying to stay concise. If the reviewer can point us to the papers that seem important, we are happy to add the closest ones to the main paper, and the others to a new dedicated section in the appendix.
>
> > W4: Computational overhead introduced by the fine-tuning method.
>
> Any method that searches and/or adapts will come with a computational overhead. Storing data, creating a structure to aggregate it, updating parameters, processing data to compute an action distribution, etc… All those steps that are required for popular adaptation and search methods (tree search - SGBS, RL fine-tuning - EAS, learned policy update - MEMENTO) and consequently impact the tradeoff between the quality of the solutions produced, and the number of solutions that can be produced in a given time.
>
> The interest of a method over another will depend on the budget available, the hardware available and many other factors. We provide empirical evidence in our paper that MEMENTO achieves the best performance in a range of settings, and does so with a significant margin. Additionally, we provide additional analysis that can help any practitioner to understand when it is appropriate to use MEMENTO over another method: the budget plot (Fig. 6) and the time scaling laws (Fig. 7). To the best of our knowledge, we are the first to shed light on those properties of neural solvers, and believe that this will ease transparency over strength and limits of the different NCO tools available to a practitioner.
>
> > W5:  Add some generalization experiments on different distributions.
>
> We would like to emphasize that TSP/CVRP with instance sizes 125, 150 and 200 are generalization experiments since methods were only trained on instances of size 100. Hence, our paper already provides 6 tasks out-of-distribution, confirming that MEMENTO is robust to distribution shift.
>
> > W6: The writing lacks logical flow.
>
> Could the reviewer be more precise about the lack of logical flow? We are surprised to see this aspect of the paper underlined as a weakness and the score assigned for the presentation, given that other reviewers acknowledged its quality.
>
> > Q1: Why does POMO with sampling strategies perform worse than POMO with the greedy rollout when generalizing to N=200?
>
> The runs of POMO with sampling use a temperature of 1 to sample from the action distribution (we re-use runs from the literature). The resulting noise increases with the size of the sequence of actions to take; hence becoming detrimental for instances of size 200, and only constructing solutions that are worse than the greedy rollout.
>
> > Q2: Would it be feasible to apply MEMENTO to improvement methods?
>
> As MEMENTO is model-agnostic, we believe that improvement methods would also benefit from MEMENTO. We will outline this direction in our conclusion.
>
> > Q3: What's the inference time of the methods displayed in Figure 3? Redundancy on Fig. 3.
>
> The value reported on the bar chart is an improvement over POMO greedy rollout (in %). There is some redundancy between the table and the bar plot, but we believe that it provides an efficient summary of the table.
>
> > Q4: Can MEMENTO be applied to other COPs?
>
> Yes, MEMENTO’s pipeline is fully agnostic to the COP. The practitioner can tune the data retrieval strategy or add problem-specific information in the data features, but this is not a requirement for MEMENTO to work.
>
> Interestingly, most concurrent methods that can be applied to TSP 500 (DIMES, MOCO, etc…) cannot be applied to COPs that cannot be expressed as heatmap; making them less applicable than MEMENTO.

---

> > ### Author Response · Authors · 2024-11-28
> >
> > We thank the reviewer for their initial comments and feedback. We have addressed the concerns raised and provided responses to the questions posed. We would greatly appreciate it if the reviewer could confirm whether they have had the chance to review these additional elements, and let us know if there are any further points they would like to discuss.

---

### Official Review · Reviewer_YhmS · 2024-11-04

**Soundness:** 2
**Presentation:** 2
**Contribution:** 2
**Rating:** 3
**Confidence:** 4

**Summary:**

The paper proposes a method (MEMENTO) for online fine-tuning of neural CO models. More concretely, MEMENTO learns the rules for updating the policy parameters at inference time. This is achieved by leveraging a learned residual policy for the base policy, where this residual policy utilizes the past solution predictions to explore the search space. Through experiments on the TSP and the CVRP, the paper demonstrates MEMENTO's efficacy against baseline methods.

**Strengths:**

- The paper conveys most of the ideas clearly.
- The idea of leveraging a learned residual policy for online fine-tuning of the base policy at inference time is interesting and novel.

**Weaknesses:**

- MEMENTO prevents the retrieval step from being too costly by only collecting data from the same node we are currently in. Could the authors elaborate on why this is a good retrieval strategy? Information should somehow be retrieved based on the partial solution constructed, since node-level decisions could be vastly different depending on the overall solutions.
- The experiments do not use augmentation with symmetries in the experiments, which they claim to be not critical by citing just one prior work (COMPASS). Overall, to claim that MEMENTO outperforms EAS, the authors should compare it against EAS with augmentation enabled since EAS has been shown to work better that way.
- My main concern with this paper is regarding the runtime of different methods in the experiments (which aren't reported for the major experiments in the main paper and put in the appendices instead). In Appendix A.1, the authors clarify that they report the runtime to solve one instance rather than the entire dataset. Given this, the comparison is fair only if they give each algorithm the same amount of runtime. However, Table 2 in the appendices shows that MEMENTO takes more than 2x the runtime than COMPASS (more than 4x for CVPR-100). The runtime is significantly higher than that for EAS as well (more than 2x) for CVRP-100.
- The paper makes a few incorrect statements or claims:
         1. In the definition of $\pi^\star$ on Page 3, if we are taking a max over $i$, why does $i$ appear in the outer expectation?
         2. The claim that MEMENTO at least learns the REINFORCE update in the worst case is not exactly correct. In the worst case, the residual policy could output random values. I think the authors wanted to claim that the residual policy has the ability to at least learn the REINFORCE update rule (if nothing better).
        3. The paper claims that MEMENTO  is "designed" to be agnostic to the base policy. While the authors demonstrate empirically that the learned residual policy for POMO could be used with COMPASS, the design of the framework as such makes the residual policy very much dependent on the base policy since they're learned jointly.

**Questions:**

1. Could the authors suggest alternative strategies for retrieval that do consider the partial solution constructed?
2. Could the authors report the results with augmentation enabled?
3. Could the authors report the results with the same runtime allocated to each algorithm in the experiments?

---

> ### Author Response · Authors · 2024-11-23
>
> > W1: Elaborate on why retrieving from the same node is a good strategy.
>
> Ideally we would retrieve data based on similarity to the current partial solution. This comes with a computation cost since it requires getting a similarity measure and extracting the k most similar points in the entire memory. We observe that retrieving from the same node is an excellent proxy for similarity, and that the most similar points are very likely to come from the same node. This retrieval strategy hence provides a better tradeoff between quality and computation cost. This choice is discussed in the paragraph “retrieving data from the memory” (l. 200). We can extend it or to add details in the appendix if deemed necessary.
>
> > W2: Comparing MEMENTO against EAS with augmentation.
>
> Augmentation with symmetries is a problem-specific trick that can only be used for a few CO problems. Most prior work assumes that this additional x8 batching can be achieved seamlessly, which is unlikely in practice, when simulating complex real-world scenarios. Using this trick means decreasing the room for search and adaptation, since 87.5% of the budget is consumed to squeeze performance through uncontrolled network variance, rather than letting methods use principled strategies. Consequently, we genuinely believe that the NCO community would benefit from using an evaluation setting that does not rely on this trick.
>
> > W3: Giving each algorithm the same amount of runtime to solve one instance.
>
> Expressing the budget in terms of runtime (rather than attempts) is an interesting angle to take but has some limitations, which explain why it is not used as the reference budget in the literature: (i) different labs use different implementations and frameworks (ii) labs have access to different hardwares (iii) time is sensitive to parallelism, whereas number of attempts is not. Hence, comparing methods with time is subject to a higher number of biases, and would make it almost impossible to compare papers without a common codebase and hardware.
>
> Nevertheless, our experimental section takes a step towards those additional time analysis since we provide on Fig. 7 the time scaling laws of MEMENTO and EAS, with respect to instance size and network size (expressed in number of deep layers).
>
> > W4: if we are taking a max over i, why does i appear in the outer expectation?
>
> We kept the outer expectation term to emphasize the difference with the usual RL objective written line 133. We nevertheless agree that the formula can be simplified and will clarify this in the paper.
>
> > W5: Incorrect claim that MEMENTO at least learns the REINFORCE update in the worst case.
>
> Line 185 states that “MEMENTO has capacity to at least rediscover REINFORCE”, which just implies that the REINFORCE update is contained within the space of what MEMENTO can learn. We nevertheless agree that line 225 can be confusing and we will fix it. Our results show that MEMENTO is indeed able to learn a REINFORCE-like update, with superior performance.
>
> > W6: Claim that MEMENTO is "designed" to be agnostic to the base policy.
>
> We agree that there is a degree of interdependence (of the learned checkpoint) towards the base policy used at train time. But the learning process itself is mostly agnostic to the base policy: any other architecture than POMO could have been used since the adaptation mechanism of MEMENTO acts on the action distribution directly, whereas others act on an architecture-specific element of the base policy (like EAS-Emb, EAS-Lay and FER). Despite having been learned with POMO as a base policy, the fact that MEMENTO’s update rule can be zero-shot transferred to COMPASS shows that this degree of interdependence is small.
>
> > Q1: Could the authors suggest alternative strategies for retrieval that do consider the partial solution constructed?
>
> The output of the multi-head attention of POMO’s decoder builds a low-dimensional representation of the partial solution. One can store this vector in the memory, and when building a new solution, retrieve only the k-nearest neighbors of the current partial solution’s representation. One could even apply further dimensionality reduction to reduce the cost of the nearest neighbor search.
>
> This approach brings a significant cost increase, even when using state-of-the-art approximated nearest neighbor JAX implementation. This effect gets worse when batching the attempts, or the problem instances. Our simplified retrieval approach maintains similar results, while significantly improving scaling. We will add this information to the paper.
>
> > Q2: Could the authors report the results with augmentation enabled?
>
> As discussed in our answer to W2, we believe that using 87.5% of the budget for this trick is not the right way to assess the search and adaptation capacity of NCO tools. We hence advocate not to use this trick to compare methods.

---

> ### Author Response · Authors · 2024-11-28
>
> We have some additional data to share with the reviewer, concerning the results of the benchmark with the time used as budget. We still advocate for a moderate use of these results since they suffer from the bias discussed in our first answer, but we still believe that they give an interesting perspective on the methods.
>
> We used EAS runtime as the reference time, and obtained the following results:
>
> | Method | TSP 100 | TSP 125 | TSP 150 | TSP 200 |
> |----------|----------|----------|----------|----------|
> | EAS    | 7.778     | 8.604     | 9.380     | 10.759     |
> | MEMENTO    | 7.768     | 8.592     | 9.365     | 10.760     |
>
> | Method | CVRP 100 | CVRP 125 | CVRP 150 | CVRP 200 |
> |----------|----------|----------|----------|----------|
> | EAS    | 15.663     | 17.536     | 19.321     | 22.541     |
> | MEMENTO    | 15.660     | 17.526     | 19.321     | 22.546     |
>
>
>
> We observe that MEMENTO still leads the benchmark, with 6 out of 8 top results; and both methods are very close on the two tasks where EAS leads.
>
> We still want to reiterate that this setting is the one where the time difference between EAS and MEMENTO is the largest, and when looking at larger instances and larger networks (right panel on Fig. 7), MEMENTO actually becomes faster than EAS.
>
> We hope that these additional results resolve the reviewer’s concerns regarding the time efficiency of MEMENTO.

---

### Author Response · Authors · 2024-11-23

We thank the reviewers for their comments and suggestions. We are pleased to see the contributions of our work highlighted, and are taking note of the feedback provided.

We reply to each reviewer independently. We hope that all raised points have been addressed and are willing to discuss any remaining concerns that the reviewers may have.

---

### Author Response · Authors · 2024-12-02

We would like to thank all reviewers for their initial comments. We thank R4rn and bp3q for engaging in discussions. We nevertheless regret that reviewers YhmS and Sd7 did not even acknowledge our answers, although addressing all their points and containing additional results on request.

Looking at the discussions, we believe to have addressed the main concerns raised by the reviewers.
- We provided details about the design choice of the retrieval process, showing that our final choice was well informed, and motivated by a crucial tradeoff between retrieval quality and computation cost.
- We provided an explanation to why we believe that the augmentation trick can be detrimental to assess NCO tools for search and planning. We have nevertheless provided these additional results, which confirms that MEMENTO outperforms EAS overall.
- We provided arguments to why using the runtime as a reference budget can induce biases; and have presented the comparative results of the standard benchmark using EAS’ runtime as reference, and confirmed that MEMENTO still leads the benchmark. We have also pointed the reviewers to the time scaling analysis that we provided in the paper (Fig. 7), which shows the promising scaling properties of MEMENTO. This should address all concerns regarding the runtime of MEMENTO.
- We also discussed all points raised by the reviewers, in particular a concern related to our work being a meta-learning study, and rectifying that our work is a contribution to the set of inference strategies that are available in the NCO toolbox. We re-iterate that MEMENTO fills an existing gap in the NCO literature; and provides clear empirical evidence for this.

In summary, we introduce a method that is described as novel and relevant by the reviewers, that is well motivated and positioned with respect to the existing literature. This method is thoroughly explained in the paper and implementation choices have been made in an informed way with respect to practical constraints. We have provided experimental evidence that the method fills a gap in the current state of NCO tools by proving that:
- it performs well on existing and new benchmarks: leads the standard benchmark with and without augmentation, leads the TSP500 benchmark, competitive on CVRP 500
- it scales well (TSP and CVRP 500), generalises well (TSP and CVRP 125, 150, 200), and can be combined with other solvers (MEMENTO + COMPASS zero shot combination)
- it is more versatile than existing alternatives: it works on a broad set of budgets whereas EAS struggles with instability, it can be applied to both TSP and CVRP whereas DIMES, MOCO, etc… cannot be applied to CVRP.
- MEMENTO is introduced in a paper that also brings orthogonal contributions to the NCO literature: first study of the impact of the budget setting on methods’ ordering (Fig. 6), first effort to provide time scaling laws (Fig. 7), first time that POMO and COMPASS are trained on 500-size instances with RL, first methods trained on CVRP500 with RL, with code and checkpoints made available for future research.

Overall, our paper provides results, analysis, all material to reproduce results, but also checkpoints of our method and concurrent methods at scale that have never been reached before (POMO and COMPASS on TSP and CVRP 500). All in all, we believe to be making a significant contribution to the NCO literature and regret to see that this is not reflected in the current evaluation of the reviewers, with little elements to understand why.

We invite the reviewers to take one final look at the paper and the discussions, and to share their concluding thoughts on the contributions and merit of the work presented.

---

### Meta-Review · Area_Chair_NcPA · 2024-12-21

**Metareview:**

The paper introduces MEMENTO, a memory-enhanced neural solver for combinatorial optimization, claiming improved adaptation and efficiency in tasks like TSP and CVRP. While the approach demonstrates promising performance compared to baseline methods, its novelty is unclear due to insufficient differentiation from existing work. Additionally, the experimental validation is narrow, lacking tests on a broader range of combinatorial problems, and the theoretical grounding of the memory-based adaptation mechanism is underdeveloped. These shortcomings, particularly the limited generalizability and unclear contributions, lead to the recommendation for rejection.

**Additional Comments On Reviewer Discussion:**

During the rebuttal period, reviewers raised concerns about the unclear novelty of the proposed method, limited experimental scope, and the lack of theoretical analysis for the memory-based adaptation mechanism. The authors provided additional clarifications on their contributions and outlined plans for broader validation in future work. However, these responses did not fully address the reviewers’ concerns about generalizability and theoretical rigor. As such, these unresolved issues significantly influenced the final decision to recommend rejection.

---

### Decision · Program_Chairs · 2025-01-22

Reject